# The vaginal microbiota associates with the regression of untreated cervical intraepithelial neoplasia 2 lesions

Anita Mitra [1,2], David A. MacIntyre [1,3], George Ntritsos[4], Ann Smith[5], Konstantinos K. Tsilidis[4,6], Julian R. Marchesi[3,7,8], Phillip R. Bennett[1,2,3], Anna-Barbara Moscicki[9,10] & Maria Kyrgiou [1,2,10 ✉]

Emerging evidence suggests associations between the vaginal microbiota (VMB) composition, human papillomavirus (HPV) infection, and cervical intraepithelial neoplasia (CIN); however, causal inference remains uncertain. Here, we use bacterial DNA sequencing from serially collected vaginal samples from a cohort of 87 adolescent and young women aged 16–26 years with histologically confirmed, untreated CIN2 lesions to determine whether VMB composition affects rates of regression over 24 months. We show that women with a *Lactobacillus*-dominant microbiome at baseline are more likely to have regressive disease at 12 months. *Lactobacillus* spp. depletion and presence of specific anaerobic taxa including *Megasphaera*, *Prevotella timonensis* and *Gardnerella vaginalis* are associated with CIN2 persistence and slower regression. These findings suggest that VMB composition may be a future useful biomarker in predicting disease outcome and tailoring surveillance, whilst it may offer rational targets for the development of new prevention and treatment strategies.

[1] Institute for Reproductive and Developmental Biology, Imperial College, Hammersmith Hospital Campus, London W12 0NN, UK. [2] Queen Charlotte's and Chelsea Hospital and Hammersmith Hospital, Imperial College Healthcare NHS Trust, London W12 OHS, UK. [3] March of Dimes Prematurity Research Centre, Imperial College London, London W12 0NN, UK. [4] Department of Hygiene and Epidemiology, University of Ioannina School of Medicine, Ioannina, Greece. [5] Division of Population Medicine, Cardiff University, Heath Park, Cardiff CF14 4YS, UK. [6] Department of Epidemiology and Biostatistics, School of Public Health, Imperial College London, London, UK. [7] School of Biosciences, Cardiff University, Cardiff CF10 3AX, UK. [8] Division of Integrative Systems Medicine and Digestive Disease, St. Mary's Hospital, Imperial College London, London W2 1NY, UK. [9] Ronald Reagan UCLA Medical Center, UCLA Mattel Children's Hospital, Santa Monica, CA, USA. [10] These authors jointly supervised this work: Anna-Barbara Moscicki, Maria Kyrgiou. ✉email: m.kyrgiou@imperial.ac.uk

Persistent infection with high-risk human papillomavirus (hrHPV) is causally associated with the development of invasive cervical cancer[1]. HPV infection is common and the lifetime risk of acquiring such an infection exceeds 80%[2]. The majority of these infections however are cleared spontaneously through an incompletely understood immune response[3]. A fraction of women with HPV persistence go on to develop the pre-invasive precursor, high-grade cervical intraepithelial neoplasia (CIN2 & 3)[4]. Cervical screening programmes are aimed at secondary prevention of cervical cancer through the ability to detect, surveil and treat CIN when necessary. Traditionally, histological diagnosis of CIN2+ has been used as the cut-off to proceed to treatment, whilst low-grade CIN (CIN1) is believed to be a histological diagnosis of benign viral replication[5].

CIN2 is often regarded as a heterogeneous disorder that can be caused by both low- and high-risk HPV (hrHPV) subtypes with various carcinogenic potential[6,7]. Despite current recommendations to treat histologically confirmed CIN2 lesions, immediate surgical management is controversial due to the high rates of regression cited by observational studies and adverse reproductive sequelae of local treatment, specifically in younger women[8,9]. Moscicki et al.[10] reported a 68% rate of spontaneous regression in 95 adolescent and young adult women with histologically confirmed CIN2 that were conservatively managed at 4-month intervals. A recent meta-analysis of 36 studies that enroled 3160 women reported a 50% rate of regression at 2 years and almost 60% when this was restricted to women under 30 years of age[8].

Emerging evidence leads us to conclude that vaginal microbiota (VMB) composition varies in women with hrHPV infections and high-grade CIN[11–15]. We previously reported that increased CIN disease severity is associated with decreasing relative abundance of *Lactobacillus* spp.[14], however, the cross-sectional nature of these datasets did not permit exploration on the impact that that VMB composition may have on clinical outcome of CIN and HPV infection clearance[11–14,16–20]. In an earlier study by Brotman et al.[11] serial sampling of women over the course of 16 weeks suggested that *Lactobacillus gasseri*-dominant communities may promote clearance of acute HPV infection. Although limited by statistical power, the study highlighted the potential utility of longitudinal profiling to examine temporal relationships between VMB and HPV infection. Furthermore, recent studies have begun to examine the impact of the VMB, HPV and cellular change on the metabolic profile, which promotes many of the inflammatory and metabolic mechanisms necessary for persistent viral infection and carcinogenesis[21,22].

In this prospective longitudinal study of historically collected samples, we investigate the vaginal microbiota composition in a cohort of non-pregnant adolescent and young adult women aged 16–26 years, with histologically proven CIN2 managed conservatively over a 24-month period. The objective of the study is to examine temporal relationships between VMB and the natural history of CIN2 and determine whether VMB composition assessed at baseline predicts outcomes at 12 and 24 months. We show that women with a *Lactobacillus*-dominant microbiome at baseline are more likely to have regressive disease at 12 months. *Lactobacillus* spp. depletion and presence of specific anaerobic taxa including *Megasphaera*, *Prevotella timonensis* and *Gardnerella vaginalis* are associated with CIN2 persistence and slower regression. Our findings suggest that VMB composition may be a future useful biomarker in predicting disease outcome and tailoring surveillance, and in addition may provide rational strategies for the development of targeted prevention and treatment methods.

## Results

### Patient cohort, characteristics and outcomes

Ninety-five women with histologically confirmed CIN2 were recruited. Eight women with missing baseline samples were excluded, giving a total of 87 women and 573 samples included for the final analysis. The mean follow-up period was 27.4 months (range 5.0–46.8 months). The mean number of biopsies during follow-up period due to clinical indications was 1 (range 0–6). An exit biopsy was offered to all patients who attended their final visit; 58 women (67%) consented to have a biopsy. Of the 87 women who entered the study with histologically confirmed CIN2, 42 had regressed by 12 months (48.3%) and the remaining 45 were classified as non-regressors (51.7%). Nine patients were subsequently lost to follow-up before they regressed and were included as 'non-regression' at 24 months (Fig. 1). At 24 months, 63 women had regressed (72.4%) and 24 had not (27.6%). Of the non-regressors, only a small number progressed to CIN3; one patient by 12 months, and a total of five by 24 months.

Patient characteristics at the baseline visit are detailed in Table 1. The mean age was 20.5 (16.0–26.5), and the mean number of sexual partners was 8 (refs. [1–35]). One in four women were smokers (24/87, 27.6%) and had previously been infected with *C. trachomatis* (22/87, 25.3%). Almost half of them had been previously pregnant (40/87, 46.0%), whereas 15% had ever practiced anal intercourse (13/87). HPV status was known for 85 patients at time of CIN2 diagnosis (97.7%), of whom 6 were HPV negative (6/85, 6.9%). Of the remaining 79 patients who were HPV positive, 73 were positive for at least one high-risk subtype (73/79, 85.8%). Thirty patients were HPV16 positive (30/79, 37.9%) and eight were infected with HPV18 (8/79, 10.1%). Twenty-eight women were infected with a single HPV subtypes (28/79, 35.4%), and fifty-one were infected with multiple HPV subtypes (51/79, 64.6%). There was no statistically significant difference in these characteristics according to regression or non-regression (Table 1).

### Baseline vaginal microbiota composition and disease outcomes

In total 2,627,778 reads were obtained from 573 samples with an average number of reads per sample of 4586 and the median read length of 370 bp after bar code removal. Following removal of singletons and rare OTUs, a total of 160 taxa were identified in the vaginal microbiota of the study cohort. To avoid sequencing bias, operational taxonomic units (OTUs) were randomly sub-sampled to the lowest read count of 885, providing a minimum coverage of 98.7% for all samples. The top 20 taxa accounted for 96.6% of the total sequence reads, and therefore further analysis was restricted to the top 20 taxa, with the remaining 140 taxa denoted as 'other'. Hierarchical clustering of genus level data for the whole cohort identified two major groups; those with ≥81.6% Lactobacillus content which we categorised as *Lactobacillus*-dominant and those with <54.2% *Lactobacillus* content, categorised as *Lactobacillus*-depleted communities, and these were observed in 65.5% (57/87) and 34.5% (30/87) of samples at baseline, respectively (Fig. 2). Similar analyses were performed at species level with hierarchical clustering analysis identifying three of the previously described CST's; CST I classified as ≥54.4% *Lactobacillus crispatus* content, CST III as ≥63.3% *Lactobacillus iners* content. Anything with <42.9% *Lactobacillus* spp. content was classified as CST IV. CST III (*L. iners*-dominant) was observed most commonly at baseline (36/87, 41.4%), followed by CST IV (*Lactobacillus* spp.-depleted, 30/87, 34.5%) and CST I (*L. crispatus*-dominant, 21/87, 24.1%) (Fig. 2). CST II and CST V, dominated by *L. gasseri* and *L. jensenii*, respectively, were not observed in any of these baseline samples, but were identified in a small number of samples at subsequent visits. VMB composition at genus or species level did not vary significantly according to HPV positivity or subtype at baseline (Supplementary Table 4).

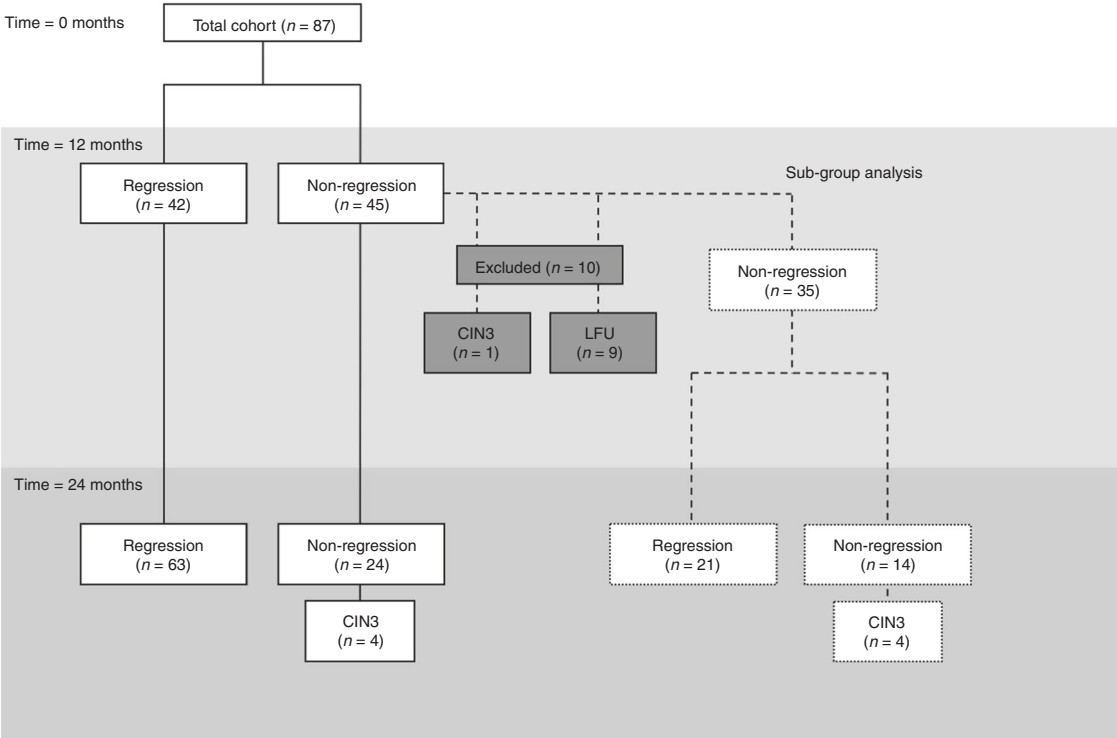

**Fig. 1 Study cohort and timepoints.** All patients entered the study at baseline with histologically confirmed CIN2. The results of follow-up histology and cytology were assessed at 12 and 24 months to determine whether the individual had regressed (at least 2× normal histology or cytology) or not (non-regression). By 12 months, 45 patients were classified as non-regressors. Of these, one patient had progressed to CIN3 and nine were subsequently lost to follow-up, leaving 35 patients for a further subgroup analysis. The vaginal microbiome composition at this 12-month follow-up visit was then regarded to be the 'new baseline', and the outcome from 12 to 24 months was then assessed. LFU lost to follow-up.

Hierarchical clustering analysis (HCA) using centroid clustering showed the VMB composition at genus level differs according to whether women showed regression of disease (determined by 2× normal cytology/histology samples), or non-regression at 12 months (adjusted odds ratio (aOR) 3.56, 95% confidence interval (CI) 1.31–9.60 ($p = 0.012$, $\chi^2$ test)), and at 24 months (aOR 2.85, 95% CI 1.03–7.92 ($p = 0.045$, $\chi^2$ test)), when adjusted for age, ethnicity, contraception, smoking, douching and HPV16 and 18 positivity (Figs. 2, 3 and Table 2). When the same analysis was performed at species level, there was a significant difference in clinical outcomes according to CST at 12 months ($p = 0.0420$, $\chi^2$ test), with CST IV at baseline being associated with a higher change of persistence at 12 (aOR 3.85, 95% CI 1.10–13.42, ($p = 0.035$, $\chi^2$ test)) and 24 months (aOR 4.25, 95% CI 0.98–18.50 ($p = 0.054$, Fisher's exact test)) compared to women with CST I at baseline (Figs. 2, 3 and Table 2). There was no significant difference in variables that may impact on VMB composition, such as contraception, douching, ethnicity, HPV status or smoking, as further shown in Table 1.

The distribution of baseline VMB composition according to clinical outcomes at 12 months is shown in Table 2 and Fig. 4. Women with a *Lactobacillus*-dominant VMB at baseline were almost twice as likely have regressed at the 12-month follow-up, compared to those with *Lactobacillus*-depleted VMB (21/30, 70.0% vs 24/57, 42.1%; aOR 3.56, 95% CI 1.31–9.60 ($p = 0.012$, $\chi^2$ test)) (Table 2, Figs. 3, 4a, b). CST IV was significantly associated with non-regression compared with women with CST I (aOR 3.85, 95% CI 1.10–13.42 ($p = 0.035$, $\chi^2$ test)). There was however no significant difference in regression rates at 12 months when comparing women with CST I and CST III (aOR 1.86, 95% CI 0.39–8.81 ($p = 0.432$, $\chi^2$ test)). Consistent with these findings, bacterial richness, as determined by the number of species

observed (Sobs) was significantly higher in women who did not regress compared to those who did ($p = 0.0105$, unpaired $t$-test), with a trend towards greater diversity, assessed using the Inverse Simpson index ($p = 0.0641$, unpaired $t$-test).

The baseline vaginal microbiota of the 45 women who had not regressed at 12 months was characterised by an increased abundance of *Megasphaera* unclassified ($p = 0.00386$, Welch's $t$-test, unadjusted), BVAB1 ($p = 0.043$, Welch's $t$-test, unadjusted) *Prevotella timonensis* ($p = 0.015$, Welch's $t$-test, unadjusted) and *Gardnerella vaginalis* ($p = 0.036$, Welch's $t$-test, unadjusted) compared to the 42 women who regressed by this timepoint, although this did not stand up to multiple test correction, likely due to sample size (Fig. 4d). LEfSe analysis identified *Lactobacillus* spp. to be predictive of regression at 12 months whereas non-regression was associated with enrichment of *Prevotella*, *Megasphaera*, BVAB1, *Sneathia* and *Atopobium* species (Fig. 5).

A similar relationship was seen between *Lactobacillus* depletion at baseline and non-regression at 24-month follow-up (aOR 2.85, 95% CI 1.03–7.92 ($p = 0.045$, $\chi^2$ test)) and with CST IV overrepresentation at baseline (aOR 4.25, 95% CI 0.98–18.50 ($p = 0.054$, $\chi^2$ test)) (Table 2, Fig. 3, Supplementary Fig. 2a, b). VMB richness at baseline was significantly greater in women with non-regression at 24 months compared to those who regressed ($p = 0.0105$, $\chi^2$ test) (Supplementary Fig. 2c, Supplementary table 2). Four species found at baseline to be significantly associated with non-regression at 12 months were again associated with non-regression at 24 months; *Prevotella timonensis* ($p = 0.03$, Welch's $t$-test, unadjusted), *Megasphaera* (unclassified) ($p = 0.033$, Welch's $t$-test, unadjusted) and *Gardnerella vaginalis* ($p = 0.037$, Welch's $t$-test, unadjusted), although this did not stand up to multiple test correction, again, likely due to sample size (Supplementary Fig. 2d).

**Table 1 Patient characteristics of 87 patients included in study cohort.**

| | Non-regression at 12 months, $n = 45$ | Regression at 12 months, $n = 42$ | All, $n = 87$ | *p*-value |
|---|---|---|---|---|
| Age, years | | | | 0.1951 |
| Mean (SD, range) | 20.1 (2.4, 16.0–24.9) | 20.8 (2.6, 16.1–26.5) | 20.5 (2.4, 16.0–26.5) | |
| Ethnicity, *n/N* (%) | | | | 0.6112 |
| Caucasian | 17/45 (37.8) | 20/42 (47.6) | 37/87 (42.5) | |
| Black | 8/45 (28.9) | 9/42 (21.4) | 17/87 (19.5) | |
| Latin | 11/45 (24.4) | 8/42 (19.0) | 19/87 (21.9) | |
| Other ethnicity | 9/45 (8.9) | 5/42 (11.9) | 14/87 (16.1) | |
| Smoking, *n/N* (%) | | | | 0.4808 |
| Current smoker | 14/45 (31.3) | 10/42 (23.8) | 24/87 (27.6) | |
| Non-smoker | 31/45 (68.9) | 32/42 (76.2) | 63/87 (72.4) | |
| Weekly alcohol use, *n/N* (%) | | | | 1.000 |
| Yes | 1/45 (2.2) | 0/42 (0) | 1/87 (95.4) | |
| No | 44/45 (97.8) | 42/42 (100.0) | 86/87 (4.6) | |
| Weekly drug use, *n/N* (%) | | | | 0.5415 |
| Yes | 5/45 (11.1) | 7/42 (16.7) | 12/87 (13.8) | |
| No | 40/45 (88.9) | 35/42 (83.3) | 75/87 (86.2) | |
| Current/previous vaginal douching, *n/N* (%) | | | | 0.6337 |
| Yes | 13/45 (28.9) | 10/42 (23.8) | 23/87 (26.4) | |
| No | 32/45 (71.1) | 32/42 (76.2) | 64/87 (73.6) | |
| Contraception, *n/N* (%) | | | | 0.6491 |
| Nil | 14/45 (31.1) | 17/42 (40.5) | 31/87 (35.6) | |
| Contraceptive injection | 8/45 (17.8) | 7/42 (16.7) | 15/87 (17.2) | |
| Other oral combined hormonal contraception | 23/45 (51.1) | 18/42 (42.8) | 41/87 (47.2) | |
| Previous pregnancy, *n/N* (%) | | | | 0.6682 |
| Yes | 22/45 (48.9) | 18/42 (42.9) | 40/87 (46.0) | |
| No | 23/45 (51.1) | 24/42 (57.1) | 47/87 (54.0) | |
| Number of sexual partners | | | | 0.5136 |
| Mean (SD, range) | 8 (8, 1–35) | 7 (6, 2–25) | 8 (7, 1–35) | |
| Current sexual practice, *n/N* (%) | | | | 0.0971 |
| Abstinence | 32/45 (71.1) | 28/42 (66.7) | 60/87 (69.0) | |
| Monogamous | 9/45 (20.0) | 4/42 (9.5) | 13/87 (15.0) | |
| Non-monogamous | 4/45 (8.9) | 10/42 (23.8) | 14/87 (16.0) | |
| History of anal intercourse, *n/N* (%) | | | | 0.7674 |
| Yes | 6/45 (13.3) | 7/42 (16.7) | 13/87 (15.0) | |
| No | 39/45 (86.7) | 35/42 (83.3) | 74/87 (85.0) | |
| History of genital infections, *n/N* (%) | | | | 0.8195 |
| *Chlamydia trachomatis* | 8/45 (17.8) | 14/42 (33.3) | 22/87 (25) | 0.1384 |
| *Neisseria gonorrhea* | 1/45 (2.2) | 3/42 (7.1) | 4/87 (5) | 0.3493 |
| *Trichomonas vaginalis* | 2/45 (4.4) | 1/42 (2.4) | 3/87 (3) | 1.000 |
| Genital warts | 5/45 (11.1) | 5/42 (11.9) | 10/87 (11) | 1.000 |
| Syphilis | 0/45 (0) | 0/42 (0) | 0/87 (0) | 1.000 |
| Bacterial vaginosis | 5/45 (11.1) | 9/42 (21.4) | 14/87 (16) | 0.2475 |
| Yeast infection | 21/45 (46.7) | 21/42 (50.0) | 42/87 (48) | 0.8313 |
| HSV | 2/45 (4.4) | 3/42 (7.1) | 5/87 (6) | 0.6693 |
| HIV | 0/45 (0) | 0/42 (0) | 0/87 (0) | 1.000 |
| HPV status, *n/N* (%) | | | | 1.000 |
| Negative | 3/45 (6.7) | 3/42 (7.1) | 6/87 (6.9) | |
| Positive | 40/45 (88.9) | 39/42 (92.9) | 79/87 (90.8) | |
| High-risk HPV status | | | | 0.4315 |
| Positive | 38/40 (95.0) | 35/39 (89.7) | 73/79 (92.4) | |
| Negative | 2/40 (5.0) | 4/39 (10.6) | 14/79 (7.6) | |
| HPV16 status | | | | 0.4826 |
| Positive | 14/38 (36.8) | 16/35 (45.7) | 30/73 (41.1) | |
| Negative | 24/38 (63.2) | 19/35 (54.3) | 43/73 (58.9) | |
| HPV18 status | | | | 0.7125 |
| Positive | 5/38 (13.2) | 3/35 (8.6) | 8/73 (11.0) | |
| Negative | 33/38 (86.8) | 32/35 (91.4) | 65/73 (89.0) | |
| Low-risk HPV status | | | | 0.4978 |
| Positive | 15/40 (37.5) | 18/39 (46.2) | 33/79 (41.7) | |
| Negative | 25/40 (62.5) | 21/39 (53.8) | 46/79 (58.3) | |
| Number of HPV subtypes | | | | 0.4823 |
| Single | 16/40 (40.0) | 12/39 (30.8) | 28/79 (35.4) | |
| Two or more | 24/40 (60.0) | 27/39 (69.2) | 51/79 (64.6) | |
| Unknown | 2/45 (4.4) | 0/42 (0) | 2/87 (2.3) | |

The *p*-values represented are two-tailed Fisher's exact (if <5 observations in any group) and $\chi^2$ tests (where >5 observations in every group).
*HIV* human immunodeficiency virus, *HSV* herpes simplex virus, *HPV* human papillomavirus, *SD* standard deviation.

Subgroup analysis of women who did or did not regress between the 12- and 24-month follow-ups was next performed (Supplementary table 1). Of the non-regressors at 12 months ($n = 45$), one progressed to CIN3 and nine were lost to follow-up, and these 10 participants were excluded, leaving 35 women with ongoing disease who were included in the subgroup analysis (Fig. 1, Supplementary Fig. 1). VMB composition at the 12-month appointment was used as the baseline comparator against which outcomes at the 24-month follow-up were measured (Table 2, Fig. 3, Supplementary Fig. 3). A trend in higher rates of *Lactobacillus* depletion and CST IV were associated with non-regression from 12 to 24 months, although this was not statistically significant (Genus level analysis (aOR 3.06, 95% CI 0.54–17.14 ($p = 0.202$, $\chi^2$ test))), species level analysis (aOR 4.94, 95% CI 0.26–94.86 ($p = 0.29$, Fisher's exact test)). There was no difference in richness or diversity according to clinical outcomes between 12 and 24 months (Supplementary Fig. 3c, Supplementary table 3). *Anaerococcus*

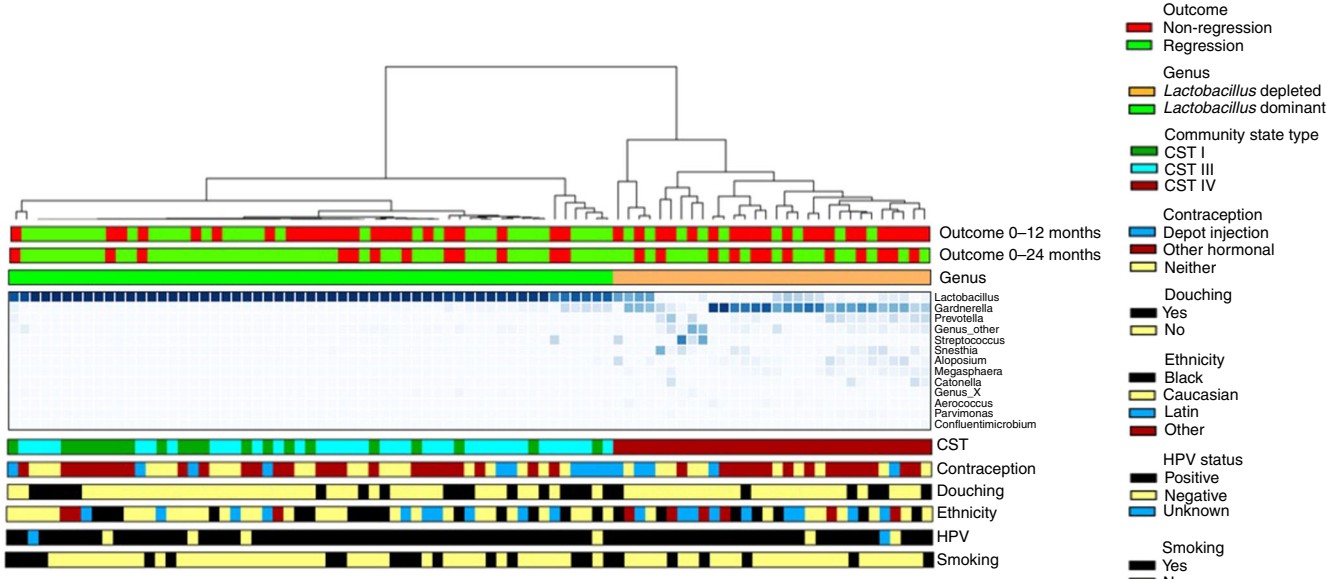

**Fig. 2 Heatmap to show vaginal microbiota composition at baseline (time = 0) according to disease outcomes between 0–12 and 0–24 month follow-up.** CST Community state type, CST I *Lactobacillus crispatus*-dominant, CST III *Lactobacillus iners*-dominant, CST IV *Lactobacillus* spp.-depleted, high diversity, HPV human papillomavirus, VMB vaginal microbiota. Top 20 taxa shown, and all remaining taxa denoted as 'other'.

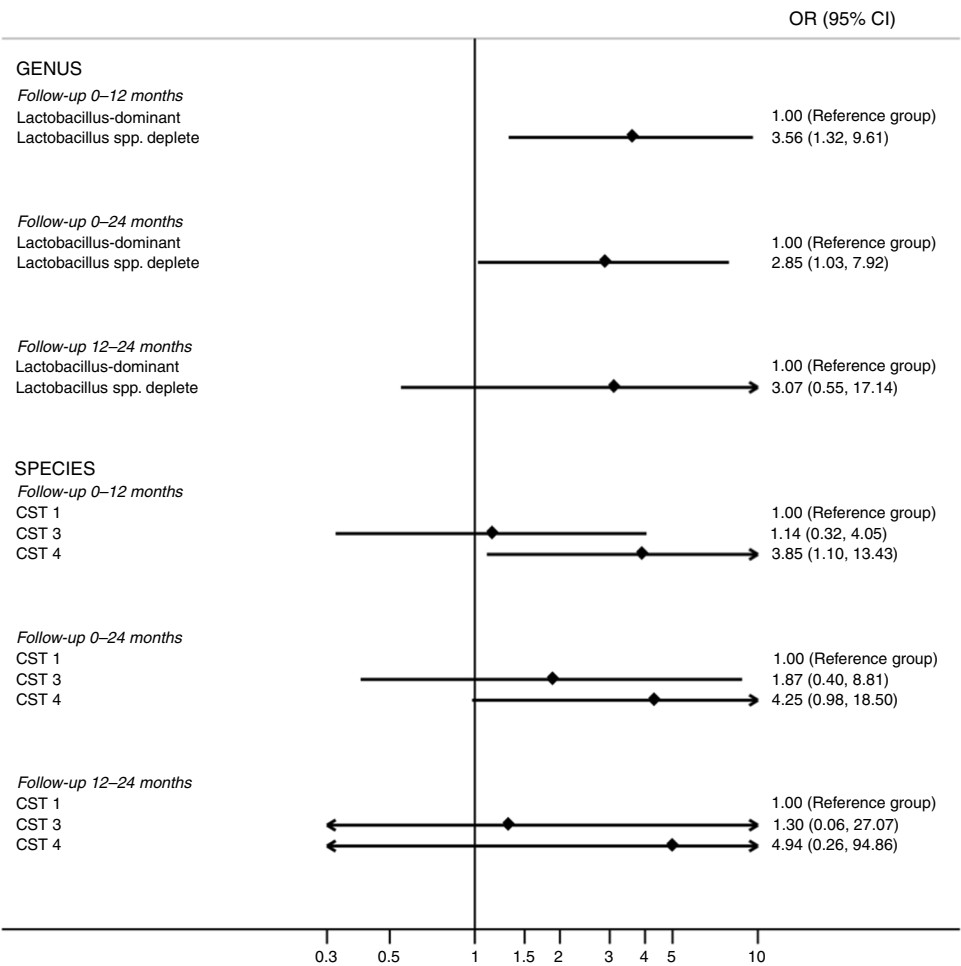

**Fig. 3 Odds ratios and 95% confidence intervals of the association between the vaginal microbiome at genera and species level with risk of CIN2 non-regression (vs regression) at different follow-up timepoints (baseline to 0–12 month; baseline to 0–24 month; 12 to 12–24 month).** Odds ratios have been adjusted for age, ethnicity, contraception, douching, smoking and HPV16 and 18 status. CI confidence interval, CST Community state type, CST I *Lactobacillus crispatus*-dominant, CST III *Lactobacillus iners*-dominant, CST IV *Lactobacillus* spp.-depleted, high diversity, OR odds ratio.

**Table 2 Outcomes according to VMB composition at 12- and 24-month follow-up at genus and species level.**

**Genus level**

|  | Regression | Non-regression | Total | Unadjusted OR, 95% CI (p-value) | Adjusted OR, 95% CI (p-value)[a] |
|---|---|---|---|---|---|
| Follow-up 0–12 months |  |  |  |  |  |
| *Lactobacillus*-dominant | 33/57 (57.9%) | 24/57 (42.1%) | 57/87 (65.5%) | REF | REF |
| *Lactobacillus* spp.-depleted | 9/30 (30.0%) | 21/30 (70.0%) | 30/87 (34.5%) | 3.21, 1.32–8.22 (0.015) | 3.56, 1.31–9.60 (0.012) |
| Total | 42/87 (48.3%) | 45/87 (51.7%) | 87/87 (100%) |  |  |
| Follow-up 0–24 months |  |  |  |  |  |
| *Lactobacillus*-dominant | 45/57 (78.9%) | 12/57 (21.1%) | 57/87 (65.5%) | REF | REF |
| *Lactobacillus* spp.-depleted | 18/30 (60.0%) | 12/30 (40.0%) | 30/87 (34.5%) | 2.50, 0.95–6.59 (0.064) | 2.85, 1.03–7.92 (0.045) |
| Total | 63/87 (72.4%) | 24/87 (27.6%) | 87/87 (100%) |  |  |
| Follow-up 12–24 months |  |  |  |  |  |
| *Lactobacillus*-dominant | 15/22 (68.2%) | 7/22 (31.8%) | 22/35 (88.0%) | REF | REF |
| *Lactobacillus* spp.-depleted | 6/13 (46.2%) | 7/13 (53.8%) | 13/35 (22.0%) | 2.50, 0.61–10.26 (0.203) | 3.06, 0.54–17.14 (0.202) |
| Total | 21/35 (60.0%) | 14/35 (40.0%) | 35/35 (100%) |  |  |

**Species level**

|  | Regression | Non-regression | Total | Unadjusted OR, 95% CI (p-value) | Adjusted OR, 95% CI (p-value)[a] |
|---|---|---|---|---|---|
| Follow-up 0–12 months |  |  |  |  |  |
| CST I | 13/21 (59.1%) | 8/21 (40.9%) | 21/87 (24.1%) | REF | REF |
| CST III | 20/36 (55.6%) | 16/36 (44.4%) | 36/87 (41.4%) | 1.30, 0.43–3.90 (0.640) | 1.14, 0.32–4.05 (0.838) |
| CST IV | 9/30 (31.0%) | 21/30 (69.0%) | 30/87 (34.5%) | 3.79, 1.17–12.30 (0.026) | 3.85, 1.10–13.42, (0.035) |
| Total | 42/87 (48.3%) | 45/87 (51.7%) | 87/87 (100%) |  |  |
| Follow-up 0–24 months |  |  |  |  |  |
| CST I | 18/21 (85.7%) | 3/21 (14.3%) | 21/87 (24.1%) | REF | REF |
| CST III | 27/36 (75.0%) | 9/36 (25.0%) | 36/87 (41.4%) | 2.00, 0.47–8.41 (0.344) | 1.86, 0.39–8.81 (0.432) |
| CST IV | 18/30 (60.0%) | 12/30 (40.0%) | 30/87 (34.5%) | 4.00, 0.96–16.61 (0.056) | 4.25, 0.98–18.50 (0.054) |
| Total | 63/87 (72.4%) | 24/87 (27.6%) | 87/87 (100%) |  |  |
| Follow-up 12–24 months |  |  |  |  |  |
| CST I | 4/5 (80.0%) | 1/5 (20.0%) | 5/35 (14.3%) | REF | REF |
| CST III | 11/16 (68.8%) | 5/16 (31.2%) | 16/35 (45.7%) | 1.82, 0.15–20.71 (0.630) | 1.30, 0.06–27.07 (0.865) |
| CST IV | 6/13 (46.2%) | 7/13 (53.8%) | 13/35 (37.1%) | 4.67, 0.40–53.95 (0.217) | 4.94, 0.26–94.86 (0.29) |
| CST V | 0/1 (0%) | 1/1 (100%) | 1/35 (2.9%) | NA[b] | NA[b] |
| Total | 21/35 (60.0%) | 14/35 (40.0%) | 35/35 (100%) |  |  |

Odds ratios (OR) are represented as unadjusted and adjusted for patient age, ethnicity, smoking, douching, contraception, HPV16 and 18 positivity, with p-values represented are two-tailed Fisher's exact (if <5 observations in any group) and $\chi^2$ tests (where >5 observations in every group).
*aOR* adjusted odds ratio, *CI* confidence Interval, *CST* community state type, CST I (*Lactobacillus crispatus*-dominant), CST III (*Lactobacillus iners*-dominant), CST IV (*Lactobacillus* spp.-depleted, high diversity), CST V (*Lactobacillus jensenii*), *NA* not applicable, *OR* odds ratio, *REF* reference population.
[a]Adjusted odds ratio and 95% confidence intervals were calculated using *Lactobacillus*-dominant (genus) or CST I (species) as a reference group and adjusted for patient age, ethnicity, smoking, douching, contraception, HPV16 and 18 positivity.
[b]It was not possible to compute the OR, 95% CI and p-value for CST V compared to CST I.

*christensenii* was significantly more abundant however, in the 12-month sample of women who did not regress at 24 months compared to those who did (*p* = 0.037, Welch's *t*-test, unadjusted) (Supplementary Fig. 3d).

**Vaginal microbiota composition and CIN2 disease clearance.**
Time to clearance of CIN according to VMB composition at genus level showed a trend towards slower clearance with a *Lactobacillus*-depleted VMB (*p* = 0.078, Log-rank test, Fig. 6). Women with CST IV at baseline had a tendency to regress slower than those with either CST I or III (*p* = 0.1864, Log-rank test) (Supplementary Fig. 4).

Sixty-three of the 87 women regressed within the 24-month follow-up period. Examination of VMB composition in the sample taken at the appointment immediately before and immediately after regression did not identify any compositional structures associated with either pre- or post-regression state, or a propensity to switch between particular compositions (Fig. 7).

Markov chain modelling was used to exploring the probability of switching CSTs within the same individual at 12 and 24 months using all available VMB composition data (Table 3,

Supplementary Fig. 5). Regression of CIN2 at 12 months was more likely to remain within CST IV (0.89), than remain in CST I (0.64) or III (0.59). Conversely, the most stable CST in non-regressors was CST I (0.75), compared to CST III (0.67) and IV (0.69). The most frequently observed transition in regressors was CST IV to CST III (0.32), compared to CST III to CST IV in non-regressors (0.28). For the 12–24-month subgroup, analysis an opposite trend was seen. CST I was the most stable VMB in regressors and CST IV most stable in non-regressors, with the most frequent transition seen between CST III to CST IV in regressors (0.29), and CST I to CST III in non-regressors (0.38) (Table 3, Supplementary Fig. 5).

**Discussion**
A frequent limitation of research investigating microbiota associations with cancer development, is the lack of longitudinal studies to help differentiate the impact of the microbiome on clinical outcomes and disease status[23]. In this study, we investigated the relationship between vaginal microbiota composition and the fate of CIN2 in a highly novel cohort of 87 young, ethnically diverse North American women. Our results suggest that

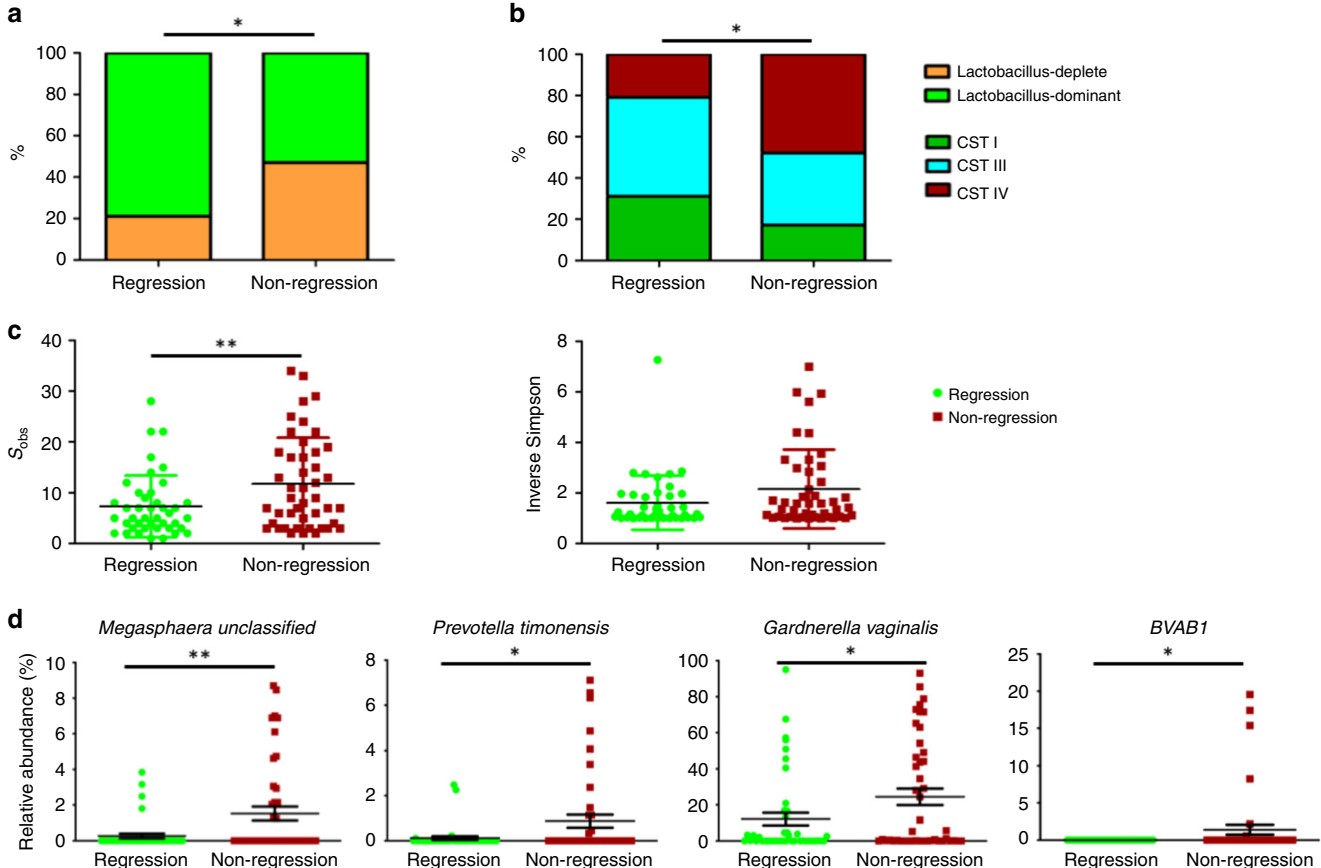

**Fig. 4 Outcomes at 12-month follow-up according to baseline VMB composition at baseline.** A *Lactobacillus* spp.-depleted VMB (**a**) and CST IV (**b**) were associated with significantly lower rates of regression compared to non-regression at 12 months. Significantly higher species richness (species observed) was seen in women who had not regressed at 12 months follow-up ($p = 0.0091$), with diversity (Inverse Simpson index) also increased but this was not significant (**c**). The baseline vaginal microbiota of women who did not regress at 12 months was characterised by an increased abundance of *Megasphaera* (unclassified) ($p = 0.0038$, q = 0.081), *Prevotella timonensis* ($p = 0.015$, q = 0.153), *Gardnerella vaginalis* ($p = 0.036$, q = 0.252) and BVAB1 ($p = 0.043$, q = 0.228) (**d**). Data are represented as percentages in **a**, **b** and **c** as mean ± standard error of mean in **d**. The *p*-values by two-tailed unpaired *t*-test in **c**, and by Welch's *t*-test in **d**. *Dots* in **c** and **d** depict individual patients. ($n = 87$, ***$p < 0.001$; **$p < 0.01$; *$p < 0.05$). Source data are supplied as a Source Data File. CST I *Lactobacillus crispatus*-dominant, CST III *Lactobacillus iners*-dominant, CST IV *Lactobacillus* spp.-depleted, high diversity.

the composition of the vaginal microbiota at the time of CIN2 diagnosis may influence the natural history of CIN2. *Lactobacillus* depletion and the presence of specific anaerobic species at the time of CIN2 diagnosis was associated with significantly lower chance of regression at 12- and 24-month follow-up. When regression did occur in these women, it tended to do so at a slower rate than in the presence of a *Lactobacillus*-dominant VMB. A similar trend was seen between VMB composition at 12 months in women with persistent disease at this time and clinical outcomes at 24 months, however reduced numbers of women maintaining disease persistence at 12 months limited the statistical power of this sub-analysis.

Temporal dynamics of VMB composition[24], can be modulated by endogenous (e.g. hormonal changes associated with menses) and exogenous factors (e.g. contraception, sexual intercourse, hygiene practices)[24–26]. It is therefore striking that VMB composition at baseline associates with CIN2 regression or non-regression 12 and 24 months later, suggesting long-term interaction between vaginal bacterial composition and CIN2 natural history, however further studies on more densely sampled women would be required to elucidate this further.

Vaginal *Lactobacillus* spp. prevent colonisation of bacterial vaginosis-associated bacterial species through maintenance of a low pH[27–30] and bacteriocin production[31–33]. An acidic environment can inhibit growth of several potentially pathogenic

species, such as *Chlamydia trachomatis*, *Neisseria gonorrhoeae* and *Gardnerella vaginalis*[27–30], yet provides optimal support for cellular metabolic function of the cervix and the vagina[34]. This feature is important for maintenance of the cervical epithelial barrier function preventing HPV access to the basal keratinocytes[35]. When strict anaerobes are able to colonise, they produce enzymes and metabolites, which may compromise this barrier, facilitating HPV entry[35]. They also act on several cellular pathways that have been associated with increased levels of proinflammatory cervical cytokines[36–39] that may enable a persistent, productive viral infection and subsequent disease development and progression[40–44] on a background of chronic inflammation, which is well-documented to promote neoplasia[45]. A recent systematic review and network meta-analysis of VMB composition and HPV status has shown that a *Lactobacillus* spp. deplete VMB is significantly associated with HPV infection compared to a CST I state (OR 4.73 (95% CI 2.06–10.86))[46]. CST IV and CST III have both previously been associated with increased acquisition and persistence of HPV infection in a 16-week longitudinal study of sexually active women who were not known to have cervical disease[11]. Other studies have associated specific bacterial taxa such as *Gardnerella*, *Atopobium* and *Megasphera* to be associated with HPV persistence[11,47], which were also highlighted as biomarkers of disease persistence in our cohort. Although BVAB1 has not previously been associated with HPV or cervical

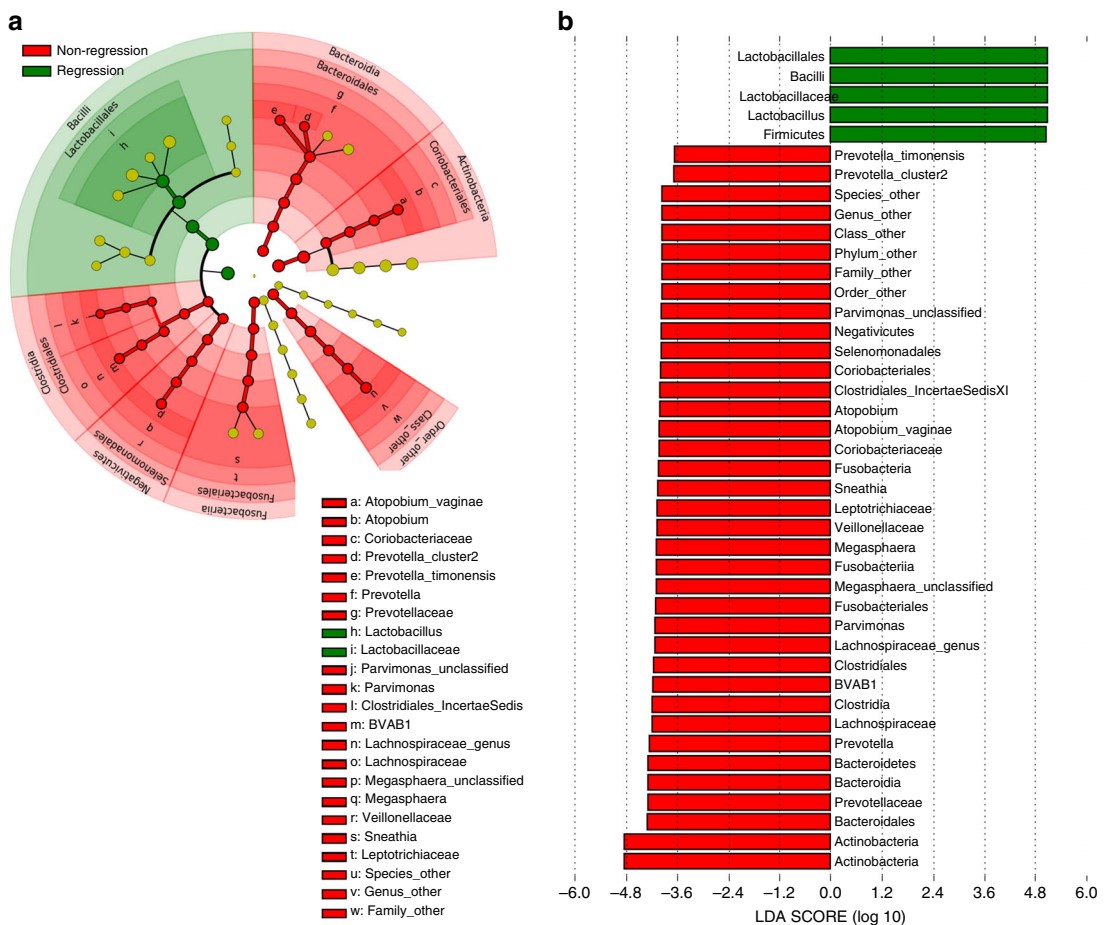

**Fig. 5 LEfSe analysis identified baseline vaginal microbiota biomarkers associated with clinical outcomes at 12 months follow-up. a** Cladogram representing taxa at all phylogenetic levels with significantly different abundance according to clinical outcome at 12-month follow-up. The size of the circle is proportional to the abundance of taxon represented. **b** Histogram of LDA scores found to differ significantly in abundance between those who regressed compared to those who did not regress. LDA linear discriminant analysis. Analysis restricted to top 20 taxa with all remaining taxa denoted as 'other'.

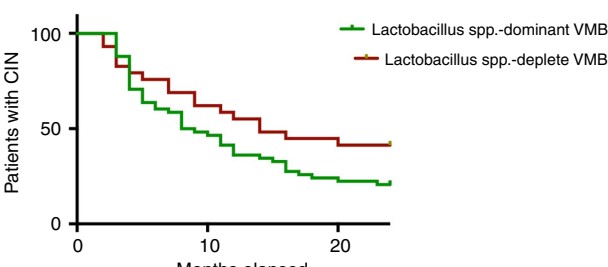

**Fig. 6 Time to clearance of CIN according to baseline VMB composition at genus level.** There was a trend towards slower disease regression in women with a *Lactobacillus* spp.-depleted VMB at baseline compared to women with a *Lactobacillus* spp.-dominant VMB at the time of CIN2 diagnosis (*p*-value=0.078, Log-rank test). Source data are supplied as a Source Data File. VMB vaginal microbiota.

disease to our knowledge, it is commonly found in a bacterial vaginosis state, and particularly associates with persistence after antibiotic treatment, and has been suggested to increase risk of HIV acquisition[48].

We did not observe any particular association between VMB composition at the time of clearance; either immediately before or immediately after, or between a switch to one particular VMB composition after clearance. However, there is likely to be a subtly

different temporality between clearance of HPV infection and clearance of any resultant CIN.

Although a control population permitting comparison of average VMB composition was not available, we observed comparatively high rates of CST III and CST IV, and lower rates of CST I than were seen in our previously described cohort of women with CIN[14]. Ethnicity has been demonstrated to influence VMB composition[49], and it is pertinent to note that this group was made up of a more ethnically diverse group compared to our previously described population which was predominantly Caucasian[14]. The higher rates of CST III and IV may also be explained by the relatively high-risk sexual behavioural characteristics of the women described in this study and as evidenced by higher rates of previous *Chlamydia trachomatis* infection, pregnancy and anal intercourse compared to the US general population at the time of recruitment[50], as well as compared to a control population without cervical disease[51]. Sexual intercourse in the absence of barrier protection is known to have an impact on the composition and stability of the vaginal microbiota, and was shown to increase the risk of *Gardnerella vaginalis* and *L. iners* colonisation in a longitudinal study of 52 sexually active women[52], and to result in reduced abundance of *L. crispatus*[53]. These data support a mechanism that puts these women at higher risk of HPV acquisition, persistence and disease development. However, we did not observe any significant differences in VMB composition according to HPV status, although women with HPV18 were more likely to have CST IV compared to those

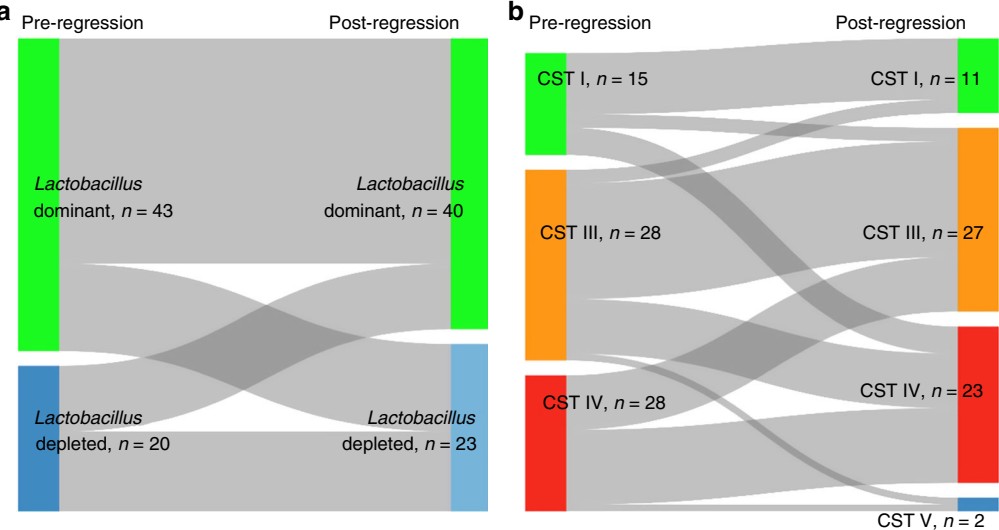

**Fig. 7 Sankey diagram of VMB composition immediately pre- and post-regression.** There was no significant difference in vaginal microbiota composition at either genus (**a**) or species level (**b**) in the sample immediately before, and the sample immediately after regression, *n* = 63. CST Community state type, CST I *Lactobacillus crispatus*-dominant, CST III *Lactobacillus iners*-dominant, CST IV *Lactobacillus* spp.-depleted, high diversity.

**Table 3 Markov modelling to assess transition probabilities of VMB dynamics among non-regressors and regressors at 0–12, 0–24 and 12–24 months.**

| From/ to | Regression | | | Non-Regression | | |
|---|---|---|---|---|---|---|
| 0–12 months | | | | | | |
| | CST I | CST III | CST IV | CST I | CST III | CST IV |
| CST I | 0.64 | 0.14 | 0.22 | 0.75 | 0.20 | 0.05 |
| CST III | 0.09 | 0.59 | 0.32 | 0.18 | 0.67 | 0.15 |
| CST IV | 0 | 0.17 | 0.83 | 0.03 | 0.28 | 0.69 |
| 0–24 months | | | | | | |
| | CST I | CST III | CST IV | CST I | CST III | CST IV |
| CST I | 0.69 | 0.19 | 0.12 | 0.75 | 0.08 | 0.17 |
| CST III | 0.12 | 0.63 | 0.25 | 0.23 | 0.58 | 0.19 |
| CST IV | 0 | 0.20 | 0.80 | 0.05 | 0.38 | 0.57 |
| 12–24 months | | | | | | |
| | CST I | CST III | CST IV | CST I | CST III | CST IV |
| CST I | 0.66 | 0.17 | 0.17 | 0.75 | 0 | 0.25 |
| CST III | 0.08 | 0.67 | 0.25 | 0.38 | 0.38 | 0.24 |
| CST IV | 0.14 | 0.29 | 0.57 | 0 | 0.14 | 0.86 |

*CST* Community state type

infected with other subtypes, yet this was not significant and may be a result of a relatively modest sample size.

Aside from environmental factors that may shift the VMB composition, there is emerging data to suggest that the host genetics also plays a role in determining microbiota composition[12,54]. Markov modelling showed that irrespective of whether CIN regressed, the VMB composition was relatively stable, which suggest that it is not the CIN itself that dictates the composition of the microbial environment. However, our results are suggestive that the VMB may drive the outcome of the disease and indicate an inverse relationship between strict anaerobes and regression. This observation is consistent with a cross-sectional cohort of women with cervical disease previously described by Oh et al.[13] who included women with LSIL or HSIL on cytology vs normal controls (defined as normal or Atypical Squamous Cells of Undetermined significance (ASCUS) cytology). They concluded that microbiota patterns, characterised by low levels of *L. crispatus* and occupied predominantly by *A. vaginae* and

secondarily by *G. vaginalis* and *L. iners*, were associated by an almost 6-fold increase in the risk of cervical LSIL/HSIL disease (higher vs lower tertile, odds ratio (OR) 5.80, 95% CI 1.73–19.4), compared to normal and thus the authors defined this as a 'risky microbial pattern'[13]. These clinical data, in addition to the in vitro studies mentioned above are clearly suggestive that *Lactobacillus* spp. have a protective role and indicate that strict anaerobes have an inflammatory impact on the cervicovaginal environment possibly enabling viral entry and facilitating persistence of HPV, which is necessary for subsequent high-grade disease, its persistence and progression. The potential interplay between the VMB and molecular pathways is further discussed in two recently published review articles[55,56].

Regression rates in this cohort were high, with 48.3% regressing by 12 months, and 72.4% by 24-month follow-up. This is much higher than many other described cohorts[57], which is likely due to the young age of the included patients, consistent with other studies in young women[7]. Our provisional findings suggest the interaction between the vaginal microbiota and natural history of CIN warrants further investigation, because it may be possible in the future to use VMB composition as a marker to identify women most at risk of persistence and progression, and even further as a therapeutic target for a more protective VMB. The use of oral probiotics therapies has been proven to modulate the composition of the VMB as a treatment for bacterial vaginosis[58,59], and are a worthwhile avenue to explore for women with a *Lactobacillus*-depleted VMB, in light of our findings that suggest the VMB composition remains stable even after clearance of CIN, because this bacterial population could put them at risk of recurrence and other adverse health outcomes associated with *Lactobacillus* depletion including HIV and STI acquisition, and pregnancy complications such as preterm rupture of membranes and preterm birth[60–63].

There are several potential confounders and limitations of the data presented in this study. Firstly, the act of taking a biopsy has been suggested to influence the natural history of CIN. There is some evidence to indicate that taking a biopsy may cause acute inflammation, which has been suggested by some to increase the chance of clearance[64], however, this point has been debated by others[65]. All patients had a biopsy at the initial visit because a histological diagnosis of CIN2 was a prerequisite for inclusion in

the study cohort. The number of subsequent biopsies carried out during follow-up ranged from 0 to 6, and therefore may represent a confounding factor not only because the biopsy may alter the natural history, but it also may uncover a higher number of cases of high-grade disease that could be missed by the relatively low sensitivity of cervical cytology[66,67].

Furthermore, in this analysis we defined as regressors women with two consecutive visits with negative cytology and/or negative biopsy, whilst LSIL was considered persistent disease as previously published for the same cohort[10]. This is also in line with definitions used in most reports exploring clinical outcomes in women with untreated CIN2 lesions[8]. The terms regression and persistence are not always interchangeable. Persistence often equates persistence of a specific lesion and regression refers to the absolute absence of the lesion. As our analysis focused on regression, categorising LSIL/CIN1 as persistence is more consistent with the initial design.

A further confounder to consider is that it is difficult to ascertain how long a lesion has been present. HPV16 and to a lesser extent the phylogenetically related subtypes HPV31, 33, 35, 52 and 58 are consistently associated with higher rates of persistence and progression[68], with HPV16 and 18, followed by the aforementioned high-risk non-16/18 subtypes, most commonly detected in cervical cancer cases[69]. This fact may be difficult to control for due to the logistics of how and when to recruit women. We were also unable to control for phase of the menstrual cycle or antibiotic use which could impact VMB composition[58,70]. Finally, we do not have a control cohort of disease-free women for comparison. It is clear that these women exhibit certain behavioural characteristics which could both alter their VMB structure and also disease outcomes.

We conclude that an absence of *Lactobacillus* spp. and presence of a diverse population of strict anaerobes at the time of CIN2 diagnosis at baseline is associated with a decreased probability of subsequent regression of untreated CIN2 lesions in young women at 12 and 24 months of surveillance. There are several plausible mechanisms for how this may arise, largely related to the development of a proinflammatory environment that may arise in the presence of a strict anaerobic environment compared to one dominated by *Lactobacillus* spp. These findings suggest that VMB composition could be a useful microbiological predictive marker of disease outcome in some women. This could be used for tailored surveillance and for the selection of women diagnosed with CIN2 that would benefit from treatment, whilst minimising overtreatment of lesions destined to regress and associated reproductive morbidity. Furthermore, this could help the development of VMB modulation therapeutic targets with pre- or probiotics that could be used for treatment and/or prevention. Future sufficiently powered longitudinal cohorts to assess the capacity of baseline microbiota composition to predict regression or progression are highly desirable and may enable the development of predictive models that could be used for risk stratification to guide clinical decision making.

## Methods

**Study population — inclusion and exclusion criteria**. Adolescent and young women between the ages of 16 to 26 years of age with histologically proven CIN2 at entry to the study were recruited at one of 12 clinics in Kaiser Permanente, Northern California, USA between 2002 and 2007 and managed conservatively with four-monthly monitoring, rather than immediate excisional treatment as described elsewhere[10]. Ethical approval was obtained from the Institutional Review Boards of the University of California, San Francisco, and Kaiser Permanente, Northern California. All patients gave written informed consent to participate, and to maintain anonymity only basic clinical data relating to disease outcome has been included in Supplementary Table 2 and 3. Further data is available upon request. Exclusion criteria included current pregnancy, previous cervical treatment for CIN, immunosuppression or plans to leave the area within the next three years. A detailed medical history was taken at recruitment and at subsequent visits to

include information regarding sexual and substance use practices. Women without a baseline sample were also excluded. Women were included irrespective of their ethnicity, parity, smoking habits, phase in their cycle and use of contraception.

A mandatory biopsy was performed at the first visit to confirm CIN2 for study entry. At subsequent visits, a cytology sample was collected and in addition, a colposcopy was performed. During follow-up, biopsies were taken on clinical grounds, i.e. suspicion of progression based on colposcopy. An exit biopsy was also performed in all patients who attended their final visit and gave consent regardless of clinical need. If the last visit was missed and no histology was available, cytology was used. In cases where there was no visible lesion at the exit visit, this biopsy was taken from the site of the previous CIN2. Histological classification of cervical biopsies at baseline was performed by the local histopathologist and sent to the centralised laboratory for verification by a second histopathologist. All other biopsies obtained through-out the study were sent directly to centralised laboratory.

**Examination and sample processing**. A cervical sample using a cytobrush and spatula was collected during each speculum examination and immediately placed in PreservCyt solution (Hologic, Marlborough, MA, USA) as per routine collection of a cervical smear. Samples were collected every 4–6 months during the 24 months follow-up. Cervical cytology and HPV genotyping were performed within one week on the PreservCyt fluid following sampling. The remaining PreservCyt sample was frozen at $-80\,^{\circ}\mathrm{C}$ until the time of bacterial DNA extraction. Whole genomic bacterial DNA was extracted from 500 μl of PreservCyt solution using a QIAmp DNA Mini kit (Qiagen, Venlo, Netherlands) according to manufacturer's instructions. After the cervical sample collection, standard colposcopic examinations were performed. If lesion appeared to progress, biopsies were taken.

**HPV genotyping**. HPV testing was performed using Roche LINEAR ARRAY® HPV Genotyping Test, a qualitative test to detect 37 high- and low-risk HPV genotypes (HPV-6, -11, -16, -18, -26, -31, -33, -35, -39, -40, -42, -45, -51, -52, -53, -54, -55, -56, -58, -59, -61, -62, -64, -66, -67, -68, -69, -70, -71, -72, -73, -81, -82, -83, -84 and -89)[71].

**Illumina MiSeq sequencing of 16S rRNA gene amplicons**. The V1–V2 hypervariable regions of 16S rRNA genes were amplified for sequencing using forward and reverse fusion primers. The forward primer consisted of an Illumina i5 adapter (5′-AATGATACGGCGACCACCGAGATCTACAC-3′), an 8-base-pair (bp) bar code, a primer pad (forward, 5′-TATGGTAATT-3′) and the 28F primer (5′-GAGT TTGATCNTGGCTCAG-3′). The reverse fusion primer was constructed with an Illumina i7 adapter (5′-CAAGCAGAAGACGGCATACGAGAT-3′), an 8-bp bar code, a primer pad (reverse, 5′-AGTCAGTCAG-3′) and the 388R primer (5′-TGC TGCCTCCCGTAGGAGT-3′). Sequencing was performed at RTL Genomics (Lubbock, TX, USA) using an Illumina MiSeq platform (Illumina Inc).

**16S rRNA gene sequence analysis**. Sequence data was processed in Mothur using the MiSeq SOP Pipeline[72]. Sequence reads were quality checked and normalised to the lowest number of reads. Singleton operational taxonomic units (OTUs) and OTUs <10 reads in any sample were collated into OTU_singletons and OTU_rare phylotypes respectively, to maintain normalisation and to minimise artefacts. OTUs were defined using a cut-off value of 97% and result data analysed using Vegan package within the R statistical package for assessment of microbial composition and diversity (R Development Core Team 2008). OTU taxonomies (from Phylum to Genus) were determined using the ribosomal database project (RDP) MultiClassifier script to generate the RDP taxonomy[73], whereas species level taxonomies of the OTUs were determined using the USEARCH algorithm (v.11) combined with the cultured representatives from the RDP[74] and STIRRUPS databases[75]. Alpha and beta indices were calculated from these datasets within Mothur and the Vegan package with the R environment (R Development Core Team 2008)[76].

**Statistical analysis**. Regression of CIN2 was defined as negative cytology and/or negative biopsy in two consecutive visits and no further cytological or histological abnormality during follow-up. Histology was used in preference to cytology; if not available, cytology was used. If the patient was lost to follow-up after a single negative cytology, the analysis was censored for the result of the last abnormal cytology or histology. Non-regression was used to define anyone with either (a) persistence; the continuing presence of low- or high-grade abnormal cytology and/ or CIN1 to CIN2 on histology, if available or (b) progression; biopsy-proven CIN3 at any follow-up visit.

Analysis of statistical differences between the vaginal microbiota of samples according to disease outcome was performed using the Statistical Analysis of Metagenomic Profiles (STAMP) package (v.2.1.3)[77]. Data were subjected to multivariate analysis using hierarchical clustering analysis (HCA) by centroid clustering with a density threshold of 0.75. Vaginal microbiota composition was classified initially into two groups at genus level according relative abundance of *Lactobacillus*; *Lactobacillus*-dominant or *Lactobacillus*-depleted. Species level data was then used to classify samples into groups analogous to previously described vaginal community state types (CSTs) I–V[49]. VMB composition at the baseline visit was compared at genus and species level according to whether women were classified as regressors or non-regressors at 12 and 24 months. We calculated odds

ratios (ORs) and 95% confidence intervals (95% CI) and p-values to explore significance using 3. We further performed a logistic regression model to adjust for known confounders (age, ethnicity, contraception, smoking, douching practice and HPV16 and 18 status) and calculated adjusted OR (aOR), using Fisher's exact and $\chi^2$ tests; adjusted ORs were reported in preferences to unadjusted. A further subgroup analysis was performed that included women who had not regressed 12 months into the study. Their VMB composition on the day of their 12-month follow-up appointment was considered a new baseline, and the outcome 12 months later (24 months since study enrolment) was observed according to the 12-month VMB composition, using these same analytical techniques. The analyses were performed in STATA statistical software (v.14).

Welch's t-test was used to perform compare relative abundance of specific species according to clinical outcomes. Linear discriminant analysis (LDA) effect size (LEfSe) analysis was used to identify taxa significantly overrepresented according to clinical outcome, through all taxonomic levels[78]. This analysis was performed using taxonomic relative abundance, with per-sample normalisation and default settings for alpha values (0.05) for the factorial Kruskal–Wallis test among classes and pairwise Wilcoxon test between subclasses. A logarithmic LDA score >2 was used to determine discriminative features.

Comparison of the VMB dynamics and stability among non-regressors and regressors at 12 and 24 months were analysed based on microbial CST transitions using Markov modelling[79]. Individuals were censored from this analysis once they regressed. Other statistical analyses were performed using the statistical package GraphPad Prism v.8.0.1 (GraphPad Software Inc., California, USA). A p-value less than 0.05 was considered statistically significant.

**Reporting summary**. Further information on research design is available in the Nature Research Reporting Summary linked to this article.

## Data availability
Sequence data that support the findings of this study have been deposited in the European Nucleotide Archive's (ENA) Sequence Read Archive (SRA) repository; https://www.ncbi.nlm.nih.gov/sra with the accession code PRJEB31832. Basic metadata relating to disease outcome is available in Supplementary Tables 2 and 3 to use alongside this to maintain anonymity. Further metadata is available upon request, however at the time of recruitment, we did not seek explicit permission to openly release all clinical data in a data repository. The source data underlying Figs. 4–6 and Supplementary Figs. 2–4 are provided as a Source Data file.

## Code availability
Markov modelling was performed using a custom R code, which has been deposited in the GitHub repository; https://github.com/anitamitra/Markov/tree/V1.0.

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

## Acknowledgements

This work was supported by the British Society of Colposcopy Cervical Pathology Jordan/Singer Award (P47773) (M.K.); Imperial College Healthcare Charity (P47907) (M.K., A.M.); Genesis Research Trust (P55549) (M.K.); Imperial Healthcare NHS Trust NIHR Biomedical Research Centre (P45272) (M.K.); NIHR Academic Clinical Fellowship programme (A.M.); Career Development Award from the Medical Research Council (MR/L009226/1) (D.A.M.); National Institutes of Health (R37 CA051323 and R01 CA87905)(A.-B.M.). The funders had no role in study design, data collection and analysis, decision to publish or preparation of the manuscript..

## Author contributions

The study was conceived and designed by A.M., A.-B.M. and M.K. The samples and patient data were acquired by A.-B.M., samples processed by A.M. and data analysed by A.M., D.A.M., G.N., A.S., K.T., J.R.M., A.-B.M. and M.K. The manuscript was drafted and revised critically for important intellectual content by all authors (A.M., D.A.M., G.N., A.S., K.T., J.R.M., P.R.B., A.-B.M., M.K.). All authors gave final approval of the version to be published and have contributed to the manuscript.

## Competing interests

Roche Molecular Diagnostics (Pleasanton, CA) provided supplies for HPV DNA detection. There are no awarded or filed patents pertaining to the results presented in the paper.
