## [Peer Review File · Nature Communications]

Reviewers' comments:

Reviewer #1 (Remarks to the Author):

Mitra and colleagues have demonstrated in a prospective cohort study that *Lactobacillus* depleted VMB and the presence of certain anaerobic species of bacteria at the time of diagnosis of CIN 2 in young adult women were associated with significantly lower chance of clearance at 12 months and 24 months (though statistically non-significant possibly due to the low sample size). The clearance rate was also slower in these women. The study is highly relevant and also consistent with the previous research data. The novelty of the study is in the prospective rigorous follow up of the subjects to establish the true disease status. I would highly recommend the manuscript to be accepted.

Just a few comments:

1. The knowledge of the p16 status at baseline would have allowed the readers to interpret the results better
2. The authors have included the women with subsequent LSIL/CIN 1 diagnosis on cytology/histology in the persistence group. This may attract criticism and should be defended by the authors
3. In the objective the authors have mentioned women in their reproductive age as the target (line 71). This should be corrected to young adult women between 16-26 years
4. Line 85: The authors have mentioned 'At 24 months, 63 women had regressed (72.4%) and 24 had not (27.6%)'. What happened to the LFU cases
5. Figure 1. Adding the LFU and CIN 3 cases in the flowchart will make it more easily interpretable.

Reviewer #2 (Remarks to the Author):

Mitra and colleagues present findings on the natural history of the vaginal microbiome in a cohort of 87 women to 26 years of age with untreated cervical intra-epithelial neoplasia (CIN2) followed over two years. The study investigates a large, ethnically-diverse cohort of women with untreated CIN over the course of two years. The cohort is unique and that is a major strength of the study, particularly as women are typically treated following a CIN2 diagnosis. The results showing that the 16 to 26 year-old women with CIN2 who went on to regress in 12 months had fewer observed species and were more likely to have a lactobacilli-dominated flora at baseline sampling are quite

convincing, and these findings are in line with other reports. The finding that the baseline microbiome matters more than the subsequent changes is clinically relevant and one of the most important contributions of this work to the field. It should be highlighted more prominently in the manuscript. There is an opportunity to determine whether predictive models could be built using these data to predict regression at 12 and 24 months. The authors also link the presence of *Megasphaera*, *Prevotella timonensis* and *Gardnerella vaginalis* to CIN2 persistence and highlight the finding that a lack of *Lactobacillus* dominance slower regression, but the strength of this conclusion is not as strong. While women with CST IV had a tendency to regress more slowly than those with lactobacilli CSTs (CST I or III) ($p=0.1864$) or lactobacilli dominance.

The manuscript is well written and the 16S rRNA microbiome analysis is generally well done. There are a few points outlined below that need attention. Rich clinical data is presented; however, covariates such as ethnicity, contraception and HPV subtype, are not fully integrated with the vaginal microbiome data other than as a display in Figure 1. Ethnicity/race and hormonal influences have strong correlations with the vaginal microbiome and these should be taken into account as covariates in the analyses. There may also be an opportunity to correlate additional information that likely exists from the biopsies with these data, and although perhaps beyond the scope of the current study, use these samples to integrate data on host factors with the microbiome.

1) Figures and Tables

a) Table 1.

It's great that detailed clinical data is available and reported for this cohort. Did any of the women become pregnant during the study period?

b) Figure 1 & Line 96 "Of these, one patient had progressed to CIN3 and nine were subsequently lost to follow-up,..."

On the left side of the figure at 24 months, it shows that non-regression N=24. It makes sense to include the subject who progressed to CIN3 in this group. It seems the 9 who were lost to follow up, should be in a separate box.

c) The sampling distribution of how and when the samples were collected is not represented. This would be a useful information to the reader as a supplementary figure or table.

d) Figure 2.

Much of figure's real estate is dedicated to taxa that are either very rare, present at very low abundance that are outside the scale, and thus it is not particularly informative. The authors could either change the color scale so that it is possible to distinguish detected at low abundance from absent or show a smaller subset of taxa that have higher abundance. As a minor point, CSTs might be better labeled as 'Community State Types' rather than 'Species'. The species-level data may be more informative here, particularly for the taxa featured as different between the regressors and non-regressors such as *Prevotella timonensis*.

e) Table 2

As a minor point, *Lactobacillus* should be italicized. Check throughout text.

f) Figure 3.

Panels A,B and C are very clear and are nice representations of significant findings. In panel D, mean proportion is not particularly informative. A representation that shows the underlying data structure such as a box plot would be much more informative. How many tests were performed and how was the list of taxa selected? Further analysis of consensus sequences would likely give finer resolution taxonomy.

g) Figure 4.

There are a number of taxa that come up using LefSE, but are not shown in Figure 3D. Were these tested in 3D? Do the reads that classify to Lachnospiraceae map to Lachnovaginotum genomospecies/ BVAB1.

h) Figure 5.

Nice visual representation.

i) Figure 6.

The authors might consider moving this figure to the supplement.

j) Figure 7.

The author might consider moving Supplementary Table 4 to the main text and Figure 7 to the Supplement. It's a clearer representation of the Markov chain modeling work, which is very nice. It is interesting that the *Lactobacillus* CSTs have transitions back to themselves in the non-regressors compared to the regressors.

2) Text

a) Line 63 "...however, the cross-sectional nature of these datasets prevents inferences of causal directionality between VMB, HPV infection and high-grade CIN (11-19)."

This is true, but the current study is also not set up to infer causal directionality. The authors may consider rephrasing.

b) Line 65. "In an earlier study by Brotman and co-workers..."

A number of recent studies in HPV, LEEP and CIN have been published that are not cited. Here is an incomplete list. The authors should revise to better put this work in context with the current state of the field.

Wiik J, Sengpiel V, Kyrgiou M, et al. Cervical microbiota in women with cervical intra-epithelial neoplasia, prior to and after local excisional treatment, a Norwegian cohort study. *BMC Womens Health*. 2019;19(1):30. Published 2019 Feb 6. doi:10.1186/s12905-019-0727-0

Ilhan ZE1, Łaniewski P2, Thomas N2, Roe DJ3, Chase DM4, Herbst-Kralovetz MM5. Deciphering the complex interplay between microbiota, HPV, inflammation and cancer through cervicovaginal metabolic profiling. EBioMedicine. 2019 Apr 24. pii: S2352-3964(19)30267-1. doi: 10.1016/j.ebiom.2019.04.028. [Epub ahead of print]

Łaniewski P1, Cui H2, Roe DJ2, Barnes D3,4, Goulder A1, Monk BJ3,4,5,6, Greenspan DL3,4,5, Chase DM2,3,4,5,6, Herbst-Kralovetz MM7,8,9. Features of the cervicovaginal microenvironment drive cancer biomarker signatures in patients across cervical carcinogenesis. Sci Rep. 2019 May 14;9(1):7333. doi: 10.1038/s41598-019-43849-5.

c) Line 116: In total 2,627,778 reads were obtained from 573 samples with an average number of reads 117 per sample of 4586 and the mean and median read lengths of 513 and 520 bp respectively & Line 416 Illumina MiSeq sequencing of 16S rRNA gene amplicons

This is a minor point/question. Was this 2x300? Are the reported read lengths the sum of R1 and R2 after filtering and primer removal? The V1-V2 region isn't that long.

d) Line 119: "To avoid sequencing bias, operational taxonomic units (OTUs) were randomly sub-sampled to the lowest read count of 885,..."

The total number of reads per sample for 16S rRNA amplicon analysis is quite low and samples were rarefied to 885 reads per sample. Based on the data shown in Figure 1, it seems that most samples were dominated by either Lactobacillus or Gardnerella or Prevotella and Sneathia in a few cases. Many of the 160 taxa are likely either very rare, very low abundance or both. While it makes sense for Sob and diversity measures, filtering to a smaller subset of taxa (and correcting for multiple testing) would make sense for comparing species between regressors and non-regressors. It's unclear how many the tests were performed in Figure 3D and Supplemental Figure 2D.

e) Line 255: "A frequent limitation of research investigating microbiota associations with cancer development, is the lack of longitudinal studies to help differentiate directional causality."

This is true, but causation is not established by the current study design. Rephrase.

f) Line 434: "...while species level taxonomies of the OTUs were determined using the USEARCH algorithm combined with the cultured representatives from the RDP database."

For taxa showing a difference or referred to in the text, attempts could be made to identify consensus sequences so the results can be put in better context with the existing literature. Uploading the consensus sequences as a Supplementary Table could also help with cross study comparisons.

g) Line 452: Vaginal microbiota composition was classified initially into two groups at genus level according relative abundance of Lactobacillus.

Describe how Lactobacillus dominance was defined here.

h) Line 454: Species-level data was used to classify samples into groups analogous to previously described vaginal community state types (CSTs) I-V.

Describe how CSTs were assigned here.

3) Data accessibility

a) The SRA study has been registered, but not yet released. I assume the study will be released prior to publication.

b) Was consent given to permit any of the clinical fields to be openly released with the data, either in the supplement or in the metadata associated with the sequences?

c) Will other clinical phenotype data also be made available (controlled-access) in a database such as dbGAP?

Reviewer #3 (Remarks to the Author):

The manuscript is well written, and I really enjoyed reading the paper. Microbial communities may play an important role in the natural history of HPV infection and the cervical carcinogenesis. As the authors concluded, in the future, the use of vaginal microbiota analysis might be useful for the triage of patients in evaluating the CIN disease severity and determining the treatment.

The study population of the present study was quite small but because the cohort consisted of women with only conservatively treated CIN2, the size of the cohort and longitudinal study setting is respectable. The study population included also women from various ethnical groups which is an important issue regarding vaginal microbiome studies. The background characteristics were well described and surprisingly similar between the regression and non-regression groups. However, I think some of the background variables should have been considered as confounders and adjusted after all. Additionally, the composition of vaginal microbiome was not determined according to the background variables (at least the results were not presented).

I have some minor comments to the manuscript:

Abstract:

- The age of the participants (16-26 years) should be mentioned.

Methods:

- The clinical definition of CIN2 regression should be described better.

- At which time points were the biopsies (1-6?) taken?
- The cohort has been collected between 2002 and 2007 – has the CIN diagnostics changed over time since then?
- Sequencing methods were appropriate and well described – however, the number of identified taxa (altogether 160) seems to be surprisingly high.

Results:

- Figure 1: Those 10 patients who were not included in the sub-group analysis of non-regressors (one who progressed to CIN3 and nine who lost the follow-up), should be marked to the flowchart.
- The word 'deplete' should be corrected as 'depleted' throughout the manuscript.
- Table 1:
 - o P-values should be presented in a separate column.
 - o It is not clear to me what 'other ethnicity' represents?
 - o Because the number of sexual partners is not presented as nominal variable in the table, n/N (%) is not needed.
 - o Why contraceptive injection (=progestin injection) is not included in the hormonal contraception group as well?
 - o P-values should be presented separately by the presence of each genital infection.
- Figure 2: This is very nice picture but showing bacterial species between genus and CST lines is not informative as the lines are barely visible. Again, I don't understand why depot progestin injection has not been included to the group of hormonal contraceptives. There was a typo in the figure legend: hierarchial. I think also that the statement 'there was no significant difference in variables that may impact on VMB composition etc..' should be found also in the manuscript text, not only in the figure legend.
- The sentence 'CST IV was significantly associated non-regression compared with women with CST I etc..' should be respelled. Especially numbers in parenthesis are not clearly presented in the current form.
- Figure 3 is very nice and informative. There is a small typo in the figure legend: Outcomes at 12 months follow-up (or 12-month follow-up)
- Figure 6 would be clearer, if pre- and post- prefixes were presented in the upper corner of figures A and B. Also, the prefix Lactobacillus should be added to -DEPLETED and -DOMINANT.

Discussion:

- Discussion was very coherent, logical and easy to read. The references were topical, but I am wondering if all the sited references (altogether 53) are needed.

REVIEWERS COMMENTS

Reviewer 1

Mitra and colleagues have demonstrated in a prospective cohort study that *Lactobacillus* depleted VMB and the presence of certain anaerobic species of bacteria at the time of diagnosis of CIN 2 in young adult women were associated with significantly lower chance of clearance at 12 months and 24 months (though statistically non-significant possibly due to the low sample size). The clearance rate was also slower in these women. The study is highly relevant and also consistent with the previous research data. The novelty of the study is in the prospective rigorous follow up of the subjects to establish the true disease status. I would highly recommend the manuscript to be accepted.

1. The knowledge of the p16 status at baseline would have allowed the readers to interpret the results better

Authors response – Thank you for your comment. Unfortunately, the p16 status of the histological sample at baseline of this historical patient cohort is not known and are not accessible since the study has been de-identified. It is worth noting that whilst there is increasing evidence to suggest p16 immunostaining of histology biopsies might be useful for CIN grading, this remains controversial, it is yet to be universally adopted, and issues of objectivity and reproducibility remain (Guedes et al. 2007 Int J Gynaecol Cancer).

2. The authors have included the women with subsequent LSIL/CIN 1 diagnosis on cytology/histology in the persistence group. This may attract criticism and should be defended by the authors

Authors response – The clinical outcomes of the cohort analysed have been previously published by one of the senior authors of this manuscript, Professor Anna-Barbara Moscicki (Ref: Moscicki AB, Ma Y, Wibbelsman C, Darragh TM, Powers A, Farhat S, et al. Rate of and risks for regression of cervical intraepithelial neoplasia 2 in adolescents and young women. *Obstet Gynecol.* 2010;116(6):1373-80). This is the largest ever published prospective cohort of young women with untreated CIN2. The definitions of progression, persistence and regression used for the analysis were similar to the ones used previously in the same cohort. The paper states “*Only those patients with at least two visits (baseline and at least one follow-up visit) were considered for this analysis. Definition of progression was biopsy proven CIN 3 at any visit after the baseline determination of CIN 2. Definition of regression was based on having three consecutive visits with negative cytology and negative biopsy on any of these visits, if available. If there was insufficient follow-up (ie, only one visit with negative cytology), then the analysis was censored at the last visit with abnormal cytology or histology. If the patient continued to have low-grade squamous intraepithelial lesion or high-grade or CIN 2 on histology at end of follow-up, then she was considered to be a non-regressor (ie, persistent).*”

In several papers included in our recent meta-analysis (Tainio K, *BMJ* 2018) exploring clinical outcomes in women with untreated CIN2 lesions, LSIL cytology and/or histology was classified as persistence, consistent with the definitions used in this paper (eg. Guedes *Anticancer Res* 2010 and Discacciati *Eur J Obs Gyn* 2011). Further, current UK guidelines for women with untreated CIN1 state two normal cytology/histology results are required to confirm regression and this is required to permit women to return back to routine recall (British Society of Colposcopy & Cervical Pathology (BSCCP) NHSCSP Guidelines no. 20 (3rd edition). Although similar definitions do not exist for untreated CIN2, which is far less commonly practiced, at least two negative cytology/histology results are usually required to determine regression and allow a woman to return back to routine recall. It is also widely accepted that diagnosis of CIN1 and CIN2 has substantial inter-observer variability and is difficult to discriminate. It is for this reason that novel biomarkers such as p16 immunostaining have

more recently been introduced. For these reasons, we felt that it would be clinically inappropriate to include a subsequent LSIL/CIN1 diagnosis as a regressor.

In summary, we do not feel that a woman with LSIL/CIN1 can be considered as a regressor. The terms regression and persistence are not always interchangeable since persistence often equates persistence of a specific lesion and regression refers to the absolute absence of the lesion. Since our analysis focused on regression, categorising LSIL/CIN1 as persistence is more consistent with the initial design.

We thank the reviewer for highlighting this and have now amended the discussion:

'Furthermore, in this analysis we defined as regressors women with two consecutive visits with negative cytology and/or negative biopsy, whilst LSIL was considered persistent disease as previously published for the same cohort (Moscicki AB Obstet Gynecol 2010). This is also in line with definitions used in most reports exploring clinical outcomes in women with untreated CIN2 lesions (Tainio K BMJ 2018). The terms regression and persistence are not always interchangeable. Persistence often equates persistence of a specific lesion and regression refers to the absolute absence of the lesion. Since our analysis focused on regression, categorizing LSIL/CIN1 as persistence is more consistent with the initial design.'

3. In the objective the authors have mentioned women in their reproductive age as the target (line 71). This should be corrected to young adult women between 16-26 years

Authors response –Thank you, we have amended this in the abstract, objective and method section and included 'adolescent and young women';

'In this prospective longitudinal study of historically-collected samples, we investigated the vaginal microbiota composition in a cohort of non-pregnant adolescent and young adult women aged 16-26 years, with histologically-proven CIN2 managed conservatively over a 24-month period.'

4. Line 85: The authors have mentioned 'At 24 months, 63 women had regressed (72.4%) and 24 had not (27.6%)'. What happened to the LFU cases

Authors response – Of the 45 women who had not regressed by 12 months, nine were lost to follow-up. In the 24-month analysis, these women were still included as 'non-regression', which we believe is appropriate because most were lost close to their 24-month follow-up. Again, this is in keeping with methods used in the previously aforementioned published study reporting on the outcomes of this cohort (Ref: Moscicki AB, Ma Y, Wibbelsman C, Darragh TM, Powers A, Farhat S, et al. Rate of and risks for regression of cervical intraepithelial neoplasia 2 in adolescents and young women. Obstet Gynecol. 2010;116(6):1373-80).

The methods section includes a clarification of how those lost to follow up were treated in the analysis:

'If the patient was lost to follow up after a single negative cytology, the analysis was censored for the result of the last abnormal cytology or histology.'

We have added a sentence to specify this in the results section. As the reviewer pointed out, this was not sufficiently clear:

'Nine patients were subsequently lost to follow-up before they regressed and were included as 'non-regression' at 24 months (Figure 1).'

We have also amended Figure 1 to include the LFU cases more clearly as suggested in the next comment from the reviewer.

5. Figure 1. Adding the LFU and CIN 3 cases in the flowchart will make it more easily interpretable.

Authors response – As suggested, LFU and CIN 3 cases have been added to Figure 1.

Reviewer 2

Mitra and colleagues present findings on the natural history of the vaginal microbiome in a cohort of 87 women to 26 years of age with untreated cervical intra-epithelial neoplasia (CIN2) followed over two years. The study investigates a large, ethnically-diverse cohort of women with untreated CIN over the course of two years. The cohort is unique and that is a major strength of the study, particularly as women are typically treated following a CIN2 diagnosis. The results showing that the 16 to 26 year-old women with CIN2 who went on to regress in 12 months had fewer observed species and were more likely to have a lactobacilli-dominated flora at baseline sampling are quite convincing, and these findings are in line with other reports. The finding that the baseline microbiome matters more than the subsequent changes is clinically relevant and one of the most important contributions of this work to the field. It should be highlighted more prominently in the manuscript.

Authors response – Thank you for the comment. We have now discussed this further in the manuscript to emphasise that the baseline VMB composition at 12 months could predict disease outcome and can influence future surveillance. This could also be useful in creating new microbiological targets for pre- or pro-biotics for prevention and/or treatment.

The Abstract and Conclusions have been amended as follows:

Abstract

'Women with a Lactobacillus-dominant microbiome at baseline were more likely to have regressive disease at 12 months.'

'These findings suggest that VMB composition may be a future useful biomarker in predicting disease outcome and tailoring surveillance, whilst it may offer rational targets for the development of new prevention and treatment strategies.'

Conclusion

'We conclude that an absence of Lactobacillus spp. and presence of a diverse population of strict anaerobes at the time of CIN2 diagnosis at baseline is associated with a decreased probability of subsequent regression of untreated CIN2 lesions in young women at 12 and 24 months of surveillance.'

'These findings suggest that VMB composition could be a useful microbiological predictive marker of disease outcome in some women. This could be used for tailored surveillance and for the selection of women diagnosed with CIN2 that would benefit from treatment, whilst minimising over-treatment of lesions destined to regress and associated reproductive morbidity. Furthermore, this could help the development of VMB modulation therapeutic targets with pre- or pro-biotics that could be used for treatment and/or prevention.'

There is an opportunity to determine whether predictive models could be built using these data to predict regression at 12 and 24 months. The authors also link the presence of Megasphaera, Prevotella timonensis and Gardnerella vaginalis to CIN2 persistence and highlight the finding that a lack of Lactobacillus dominance slower regression, but the strength of this conclusion is not as strong. While women with CST IV had a tendency to regress more slowly than those with lactobacilli CSTs (CSTI or III) (p=0.1864) or lactobacilli dominance.

Authors response – We agree that this is a unique longitudinal cohort of serial samples in women with untreated CIN2 lesions that will give more insights on the impact of VMB on disease outcome. As suggested, we have now included Odds Ratios (ORs) as useful statistical measures of clinical reporting of outcomes. We also included the adjusted and unadjusted ORs in Table 2 and created an additional Figure 3 summarising the ORs that would be useful to the reader.

We also agree with the reviewer that this and future larger longitudinal cohorts of women with HPV+/-CIN would provide an opportunity for future development of predictive models that should be

built with stringent statistical methodologies to include discrimination, calibration and internal or external validation. The development of such a predictive model in this cohort is severely limited by sample size. However, we have now amended the discussion to include comment on the fact that our cohort of untreated CIN2 women is a unique cohort and our findings open avenues for future research in larger longitudinal cohorts.

Details of the new analyses and edits to the results, tables, figures and discussion are presented below.

Results:

'Hierarchical clustering analysis (HCA) using centroid clustering showed the VMB composition at genus level differs according to whether women showed regression of disease (determined by 2x normal cytology/histology samples), or non-regression at 12 months (adjusted odds ratio (aOR) 3.56, 95% confidence interval (CI) 1.31 - 9.60 (p=0.012)), and at 24 months (aOR 2.85, 95% CI 1.03 -7.92 (p=0.045)), when adjusted for age, ethnicity, contraception, smoking, douching and HPV-16 & -18 positivity (Figure 2 & 3, Table 2). When the same analysis was performed at species level, there was a significant difference in clinical outcomes at 12 according to CST at 12 months (p=0.0420), with CST IV at baseline being associated with a higher change of persistence at 12 (aOR 3.85, 95% CI 1.10 – 13.42, (p=0.035)) and 24 months (aOR 4.25, 95% CI 0.98 – 18.50 (p=0.054)) compared to women with CST I at baseline (Figure 2 & 3, Table 2).'

'The distribution of baseline VMB composition according to clinical outcomes at 12 months are shown in Table 2 and Figure 4. Women with a Lactobacillus-dominant VMB at baseline were almost twice as likely have regressed at the 12-month follow-up, compared to those with Lactobacillus-depleted VMB (21/30, 70.0% vs 24/57, 42.1%; aOR 3.56, 95% CI 1.31 - 9.60 (p=0.012)) (Table 2, Figure 3, 4A & B). CST IV was significantly associated with non-regression compared with women with CST I (aOR 3.85, 95% CI 1.10 – 13.42, (p=0.035)). There was however no significant difference in regression rates at 12 months when comparing women with CST I and CST III (aOR 1.86, 95% CI 0.39 – 8.81, (p=0.432)). Consistent with these findings, bacterial richness, as determined by the number of species observed (Sobs) was significantly higher in women who did not regress compared to those who did (p=0.0105), with a trend towards greater diversity, assessed using the Inverse Simpson index (p=0.0641).'

'A similar relationship was seen between Lactobacillus depletion at baseline and non-regression at 24-month follow-up (aOR 2.85, 95% CI – 1.03-7.92 (p=0.045)) and with CST IV overrepresentation at baseline (aOR 4.25, 95% CI – 0.98-18.50 (p=0.054)) (Table 2, Figure 3, Supplementary Figure 2A & B). VMB richness at baseline was significantly greater in women with non-regression at 24 months compared to those who regressed (p=0.0105) (Supplementary Figure 2C). Four species found at baseline to be significantly associated with non-regression at 12 months were again associated with non-regression at 24 months; Prevotella timonensis (p=0.03), Megasphaera (unclassified) (p=0.033), Gardnerella (unclassified) (p=0.048), and Gardnerella vaginalis (p=0.037) (Supplementary Figure 2D).'

Table 1. Outcomes according to VMB composition at 12- and 24-month follow-up at genus and species level.

KEY – aOR: adjusted odds ratio, CI: Confidence Interval, CST: Community state type, CST I (Lactobacillus crispatus-dominant), CST III (Lactobacillus iners-dominant), CST IV (Lactobacillus spp. depleted, high diversity), CST V (Lactobacillus jensenii), NA: Not applicable, OR: odds ratio, REF: reference population.

**Adjusted odds ratio and 95% confidence intervals were calculated using Lactobacillus—dominant (genus) or CST I (Species) as a reference group and adjusted for patient age, ethnicity, smoking, douching, contraception, HPV-16 and -18 positivity. ^It was not possible to compute the OR, 95% CI & p-value for CST V compared to CST I.*

GENUS LEVEL					
Follow-up 0-12 months	Regression	Non-regression	Total	Unadjusted OR, 95% CI (p-value)	Adjusted OR, 95% CI (p-value)*
Lactobacillus-dominant	33/57 (57.9%)	24/57 (42.1%)	57/87 (65.5%)	REF	REF
Lactobacillus spp. depleted	9/30 (30.0%)	21/30 (70.0%)	30/87 (34.5%)	3.21, 1.32 – 8.22 (0.015)	3.56, 1.31 – 9.60 (0.012)
Total	42/87 (48.3%)	45/87 (51.7%)	87/87 (100%)		
Follow-up 0-24months	Regression	Non-regression	Total	Unadjusted OR, 95% CI (p-value)	Adjusted OR, 95% CI (p-value)*
Lactobacillus-dominant	45/57 (78.9%)	12/57 (21.1%)	57/87 (65.5%)	REF	REF
Lactobacillus spp. depleted	18/30 (60.0%)	12/30 (40.0%)	30/87 (34.5%)	2.50, 0.95 – 6.59 (0.064)	2.85, 1.03 – 7.92 (0.045)
Total	63/87 (72.4%)	24/87 (27.6%)	87/87 (100%)		
Follow-up 12-24 months	Regression	Non-regression	Total	Unadjusted OR, 95% CI (p-value)	Adjusted OR, 95% CI (p-value)*
Lactobacillus-dominant	15/22 (68.2%)	7/22 (31.8%)	22/35 (88.0%)	REF	REF
Lactobacillus spp. depleted	6/13 (46.2%)	7/13 (53.8%)	13/35 (22.0%)	2.50, 0.61 – 10.26 (0.203)	3.06, 0.54 – 17.14 (0.202)
Total	21/35 (60.0%)	14/35 (40.0%)	35/35 (100%)		
SPECIES LEVEL					
Follow-up 0-12 months	Regression	Non-regression	Total	Unadjusted OR, 95% CI (p-value)	Adjusted OR, 95% CI (p-value)*
CST I	13/21 (59.1%)	8/21 (40.9%)	21/87 (24.1%)	REF	REF
CST III	20/36 (55.6%)	16/36 (44.4%)	36/87 (41.4%)	1.30, 0.43 – 3.90 (0.640)	1.14, 0.32 – 4.05 (0.838)
CST IV	9/30 (31.0%)	21/30 (69.0%)	30/87 (34.5%)	3.79, 1.17 – 12.30 (0.026)	3.85, 1.10 – 13.42 (0.035)
Total	42/87 (48.3%)	45/87 (51.7%)	87/87 (100%)		
Follow-up 0-24months	Regression	Non-regression	Total	Unadjusted OR, 95% CI (p-value)	Adjusted OR, 95% CI (p-value)*
CST I	18/21 (85.7%)	3/21 (14.3%)	21/87 (24.1%)	REF	REF
CST III	27/36 (75.0%)	9/36 (25.0%)	36/87 (41.4%)	2.00, 0.47 – 8.41 (0.344)	1.86, 0.39 – 8.81 (0.432)
CST IV	18/30 (60.0%)	12/30 (40.0%)	30/87 (34.5%)	4.00, 0.96 – 16.61 (0.056)	4.25, 0.98 – 18.50 (0.054)
Total	63/87 (72.4%)	24/87 (27.6%)	87/87 (100%)		
Follow-up 12-24months	Regression	Non-regression	Total	Unadjusted OR, 95% CI (p-value)	Adjusted OR, 95% CI (p-value)*
CST I	4/5 (80.0%)	1/5 (20.0%)	5/35 (14.3%)	REF	REF
CST III	11/16 (68.8%)	5/16 (31.2%)	16/35 (45.7%)	1.82, 0.15 – 20.71 (0.630)	1.30, 0.06 – 27.07 (0.865)
CST IV	6/13 (46.2%)	7/13 (53.8%)	13/35 (37.1%)	4.67, 0.40 – 53.95 (0.217)	4.94, 0.26 – 94.86 (0.29)
CST V	0/1 (0%)	1/1 (100%)	1/35 (2.9%)	NA^	NA^
Total	21/35 (60.0%)	14/35 (40.0%)	35/35 (100%)		

Figure 3. Odds ratios and 95% confidence intervals of the association between the vaginal microbiome at genera and species level with risk of CIN2 non-regression (versus regression) at different follow-up timepoints (baseline to 0-12m; baseline to 0-24m; 12m to 12-24m). Odds ratios have been adjusted for age, ethnicity, contraception, douching, smoking and HPV-16 and -18 status. *KEY – CI: confidence interval, CST: Community state type, CST I: Lactobacillus crispatus-dominant, CST III: Lactobacillus iners-dominant), CST IV: Lactobacillus spp. depleted, high diversity, OR: odds ratio.*

Discussion:

‘Future sufficiently powered longitudinal cohorts to assess the capacity of baseline microbiota composition to predict regression or progression are highly desirable and may enable the development of predictive models that could be used for risk stratification to guide clinical decision making.’

The manuscript is well written and the 16S rRNA microbiome analysis is generally well done. There are a few points outlined below that need attention. Rich clinical data is presented; however, covariates such as ethnicity, contraception and HPV subtype, are not fully integrated with the vaginal microbiome data other than as a display in Figure 1. Ethnicity/race and hormonal influences have strong correlations with the vaginal microbiome and these should be taken into account as covariates in the analyses. There may also be an opportunity to correlate additional information that likely exists from the biopsies with these data, and although perhaps beyond the scope of the current study, use these samples to integrate data on host factors with the microbiome.

Authors response –

We agree that there are a number of confounders that may influence microbiome composition. In our manuscript, we first assessed whether there were significant differences in the proportions of potential cofounders (e.g. ethnicity, contraception) between the two compared groups as presented in Table 1. Both in this analysis, and in the original analysis of the same cohort presented by Moscicki AM Obs Gyn 2010, we found no difference between the two groups.

However, we agree with the reviewer that further statistical modelling would strengthen the paper's inference. In response to the comments raised by reviewer 1, reviewer 3 and the editorial board, we performed a multivariable logistic regression analysis and present the new adjusted p-values and odds ratios in Table 2. We adjusted for a number of covariates including age, ethnicity, contraception, smoking, douching practice and HPV-16 and -18 status. We further calculated adjusted and unadjusted Odds ratios and 95%CI as well as unadjusted and adjusted p-values. These have now been incorporated in Table and we also added Figure 3 in the manuscript.

Importantly, these results have not change our major findings and have strengthened the robustness of our conclusions and results. Amendments to the manuscript are as follows:

Methods:

VMB composition at the baseline visit was compared at genus and species level according to whether women were classified as regressors or non-regressors at 12 and 24 months. We calculated odds ratios (ORs) and 95% confidence intervals (95%CI) and p-values to explore significance using Fischer's exact and chi-square tests. We further performed a logistic regression model to adjust for known confounders (age, ethnicity, contraception, smoking, douching practice and HPV-16 and -18 status) and calculated adjusted OR (aOR); tests; adjusted ORs were reported in preferences to unadjusted. A further subgroup analysis was performed that included women who had not regressed 12 months into the study. Their VMB composition on the day of their 12-month follow-up appointment was considered a new baseline, and the outcome 12 months later (24 months since study enrollment) was observed according to the 12-month VMB composition, using these same analytical techniques. The analyses were performed in STATA statistical software v.14.

Table 2, Results, Figure 3:

The changes in Table 2, Results and the additional Figure 3 can be found in the response to an earlier comment.

1) Figures and Tables

a) Table 1.

It's great that detailed clinical data is available and reported for this cohort. Did any of the women become pregnant during the study period?

Authors response – None of the women became pregnant during the follow-up period.

b) Figure 1 & Line 96 “Of these, one patient had progressed to CIN3 and nine were subsequently lost to follow-up,...”. On the left side of the figure at 24 months, it shows that non-

regression N=24. It makes sense to include the subject who progressed to CIN3 in this group. It seems the 9 who were lost to follow up, should be in a separate box.

Authors response – We have amended Figure 1 to now represent these groups.

e) The sampling distribution of how and when the samples were collected is not represented. This would be a useful information to the reader as a supplementary figure or table.

Authors response – The technique for sample collection is similar to a routine collection of a liquid cervical cytology sample using cervical brush and spatula for sample collection. Samples were collected every 4-6 months during the 24-month follow-up period. We have amended the Methods section to clarify:

Methods

'A cervical sample using a cytobrush and spatula was collected during each speculum examination and immediately placed in PreservCyt solution (Hologic, Marlborough, MA, USA) as per routine collection. Samples were collected every 4-6 months during the 24 months follow-up. Cervical cytology and HPV genotyping were performed within one week on the PreservCyt fluid following sampling. The remaining PreservCyt sample was frozen at -80°C until the time of bacterial DNA extraction. Whole genomic bacterial DNA was extracted from 500µl of PreservCyt solution using a QIAmp DNA Mini kit (Qiagen, Venlo, Netherlands) according to manufacturer's instructions. After the cervical sample collection, standard colposcopic examinations were performed. If lesion appeared to progress, biopsies were taken.'

d) Figure 2.

Much of figure's real estate is dedicated to taxa that are either very rare, present at very low abundance that are outside the scale, and thus it is not particularly informative. The authors could either change the color scale so that it is possible to distinguish detected at low abundance from absent or show a smaller subset of taxa that have higher abundance. As a minor point, CSTs might be better labeled as 'Community State Types' rather than 'Species'. The species-level data may be more informative here, particularly for the taxa featured as different between the regressors and non-regressors such as *Prevotella timonensis*.

Authors response – In consideration of the Reviewers' comments, we have substantially altered Figure 2 by limiting the heatmap presented to the top 20 taxa, which account for 97% of the total sequence reads in the data set. Remaining taxa are represented as 'other' in the heatmap. The major findings of the paper have not changed, supporting the fact that these less frequently observed taxa are not driving the observed outcomes. We have re-labelled 'species' as 'Community State Type' in Figure 2 & Supplementary Figure 1. As both genus and species level data provide complementary information, we present both taxonomic classifications. Analysis of data using genera-level classifications improves statistical power and permits assessment of whether *Lactobacillus* spp. dominance or depletion is a contributor to observed outcomes. However, we agree that species-level data provides higher resolved information on those taxa specifically associated with clinical outcomes that could inform further studies on microbiological markers.

e) Table 2

As a minor point, *Lactobacillus* should be italicized. Check throughout text.

Authors response – Thank you, we have amended this.

f) Figure 3.

Panels A, B and C are very clear and are nice representations of significant findings. In panel D, mean proportion is not particularly informative. A representation that shows the underlying data structure such as a box plot would be much more informative. How many tests were

performed and how was the list of taxa selected? Further analysis of consensus sequences would likely give finer resolution taxonomy.

Authors response –

Based on the reviewer’s previous suggestion, we have now re-analysed the data using only the top 20 taxa. Figure 3D (which has now become Figure 4 due to the addition of another figure) is a representation of the taxa at species level that were found to differ in relative abundance between the regression and non-regression groups using Welch’s t-test. When corrected for multiple testing, the significance does not remain, which is likely due to sample size, however we have now presented corrected and uncorrected p-values in the manuscript. As suggested, we have now presented data for significantly different taxa as box plots representing relative abundance. The supplementary figures have also been changed to box plots.

At the reviewer’s suggestion, we have improved taxonomy resolution by using the STIRRUPS database, a vaginal 16S rDNA Reference Database that employs the USEARCH algorithm with a curated reference database for species-level classification. The database is comprised of a comprehensive and non-redundant database of 16S rDNA reference sequences for bacterial taxa likely to be associated with vaginal health. Using this approach we were able to classify an additional 3 further species from the Top 20 OTUs (and 10 overall). These were; OTUs: Otu006 - *Sneathia_ammii*, Otu013 as BVAB1 and Otu017 - *Prevotella_cluster2*. We have now updated the methods to clearly state how assignment of taxonomy was performed and updated figures to reflect the newly assigned OTUs.

Figure 4. Outcomes at 12-month follow-up according to baseline VMB composition at baseline
 A *Lactobacillus* spp. depleted VMB (A) and CST IV (B) were associated with significantly lower rates of regression compared to non-regression at 12 months. Significantly higher species richness (species observed) was seen in women who had not regressed at 12 months follow-up (p=0.0091), with a trend increase in diversity (Inverse Simpson index). The baseline vaginal microbiota of women

who did not regress at 12 months was characterised by an increased abundance of *Megasphaera* (unclassified)($p=0.0038$), *Prevotella timonensis* ($p=0.015$), *Gardnerella vaginalis* ($p=0.036$) and BVAB1 ($p=0.043$) (Welch's t-test) (D).

KEY - CST I: Lactobacillus crispatus-dominant, CST III: Lactobacillus iners-dominant), CST IV: Lactobacillus spp. depleted, high diversity

g) Figure 4. There are a number of taxa that come up using LefSE, but are not shown in Figure 3D. Were these tested in 3D? Do the reads that classify to Lachnospiraceae map to Lachnovaginotum genomospecies/ BVAB1.

Authors response –

Inconsistencies between taxa shown in 3D (now Figure 4D) and LefSE were primarily due to differences in the statistical procedures used. Taxa in Figure 3D were identified by comparing the means of relative abundance at species level taxonomy between regression and non-regression using an uncorrected Welch's t-test. LefSE on the other hand, uses multiple hypothesis testing and estimation of effect size at all taxonomic levels simultaneously to generate a list of differentially abundant taxa between two classes. Specifically, it first uses the non-parametric factorial Kruskal-Wallis (KW) sum-rank test to detect features with significant differential abundance with respect to the class of interest (i.e. regression versus non-regression), without correction for multiple testing. Importantly, all pairwise comparisons are required to reject the null hypothesis for detecting a microbial biomarker at a given taxonomic level, thus, no multiple testing corrections are needed (Segata et al., Genome Biology 2011) Finally, LefSE then uses linear discriminant analysis (LDA) to estimate the effect size of each differentially abundant feature.

However, following the Reviewer's suggestion, we have now limited our analyses to the top 20 most abundant taxa (accounting for >97% total reads). This has led to consistent findings between univariate analyses (see Figure 4D) and LefSE analyses, although some minor differences will remain due to the reasons outlined above.

We have also been able to improve assignments at species level (see earlier replies) and can confirm that reads for Lachnospiraceae do indeed map to BVAB1.

i) Figure 6. The authors might consider moving this figure to the supplement.

Authors response – We would prefer to include the Sankey diagrams in the main manuscript as we feel that they provide useful and easily interpretable visual representations of the changes in the microbiome composition. If the Reviewer and/or editorial board feel strongly, we would be willing to move to the supplement.

j) Figure 7. The author might consider moving Supplementary Table 4 to the main text and Figure 7 to the Supplement. It's a clearer representation of the Markov chain modeling work, which is very nice. It is interesting that the Lactobacillus CSTs have transitions back to themselves in the non-regressors compared to the regressors.

Authors response – As suggest, we have moved Supplementary Table 4 to the main text, and Figure 7 has been moved to the supplement to become Supplementary Figure 5.

2) Text

a) Line 63 "...however, the cross-sectional nature of these datasets prevents inferences of causal directionality between VMB, HPV infection and high-grade CIN (11-19)." This is true, but the current study is also not set up to infer causal directionality. The authors may consider rephrasing.

Authors response – We agree with the reviewer that causal directionality would require further mechanistic and intervention studies. This has been rephrased in the manuscript:

'however, the cross-sectional nature of these datasets did not permit exploration on the impact that that VMB composition may have on clinical outcome of CIN and HPV infection clearance.'

b) Line 65. "In an earlier study by Brotman and co-workers..."
A number of recent studies in HPV, LEEP and CIN have been published that are not cited. Here is an incomplete list. The authors should revise to better put this work in context with the current state of the field.

Wiik J, Sengpiel V, Kyrgiou M, et al. Cervical microbiota in women with cervical intra-epithelial neoplasia, prior to and after local excisional treatment, a Norwegian cohort study. *BMC Womens Health*. 2019;19(1):30. Published 2019 Feb 6. doi:10.1186/s12905-019-0727-0

Ilhan ZE1, Łaniewski P2, Thomas N2, Roe DJ3, Chase DM4, Herbst-Kralovetz MM5. Deciphering the complex interplay between microbiota, HPV, inflammation and cancer through cervicovaginal metabolic profiling. *EBioMedicine*. 2019 Apr 24. pii: S2352-3964(19)30267-1. doi: 10.1016/j.ebiom.2019.04.028. [Epub ahead of print]
Łaniewski P1, Cui H2, Roe DJ2, Barnes D3,4, Goulder A1, Monk BJ3,4,5,6, Greenspan DL3,4,5, Chase DM2,3,4,5,6, Herbst-Kralovetz MM7,8,9. Features of the cervicovaginal microenvironment drive cancer biomarker signatures in patients across cervical carcinogenesis. *Sci Rep*. 2019 May 14;9(1):7333. doi: 10.1038/s41598-019-43849-5.

Authors response – Thank you for drawing our attention to these important contributions to the field. Although we note that one of the reviewers requested the number of references to be reduced, we have included reference to these studies in the introduction as outlined below. We are happy to be guided by the editorial board as to whether the number of references are suitable or not.

Introduction:

'Emerging evidence leads us to conclude that vaginal microbiota (VMB) composition varies in women with hrHPV infections and high-grade CIN (11-15).'

'Furthermore, recent studies have begun to examine the impact of the VMB, HPV and cellular change on the metabolic profile, which promotes many of the inflammatory and metabolic mechanisms necessary for persistent viral infection and carcinogenesis (20, 21).'

c) Line 116: In total 2,627,778 reads were obtained from 573 samples with an average number of reads per sample of 4586 and the mean and median read lengths of 513 and 520 bp respectively & Line 416 Illumina MiSeq sequencing of 16S rRNA gene amplicons This is a minor point/question. Was this 2x300? Are the reported read lengths the sum of R1 and R2 after filtering and primer removal? The V1-V2 region isn't that long.

Authors response –

Thank you for highlighting this error. We have now adjusted the manuscript to report the accurate median read lengths for the sequencing data, which was indeed generated using 2x300 regions and the barcodes removed.

d) Line 119: "To avoid sequencing bias, operational taxonomic units (OTUs) were randomly sub-sampled to the lowest read count of 885,..."

The total number of reads per sample for 16S rRNA amplicon analysis is quite low and samples were rarefied to 885 reads per sample. Based on the data shown in Figure 1, it seems that most samples were dominated by either *Lactobacillus* or *Gardnerella* or *Prevotella* and *Sneathia* in a few cases. Many of the 160 taxa are likely either very rare, very low abundance or both. While it makes sense for Sob and diversity measures, filtering to a smaller subset of taxa (and correcting for multiple testing) would make sense for comparing species between regressors and non-regressors. It's unclear how many the tests were performed in Figure 3D and Supplemental Figure 2D.

Authors response – We agree with the reviewer that our data highlights the fact that generally, most samples were dominated by a small number of taxa, with remaining taxa being rare or low in relative abundance or indeed, both. This is further confirmed by examining coverage after rarefaction, which was >99% in all samples. In response to an earlier comment raised by the reviewer, we have now limited our analyses to the top 20 most abundant species, which collectively account for >97% of total sequence reads. Multiple testing was controlled for using Benjamini-Hochberg correction and we now present both uncorrected and corrected p-values in the manuscript. Importantly, these new analyses have not changed the major findings as originally presented.

e) Line 255: “A frequent limitation of research investigating microbiota associations with cancer development, is the lack of longitudinal studies to help differentiate directional causality.” This is true, but causation is not established by the current study design. Rephrase.

Authors response – This has been rephrased as follows:

‘A frequent limitation of research investigating microbiota associations with cancer development, is the lack of longitudinal studies to help differentiate the impact of the microbiome on clinical outcomes and disease status (22).’

f) Line 434: “...while species level taxonomies of the OTUs were determined using the USEARCH algorithm combined with the cultured representatives from the RDP database.” For taxa showing a difference or referred to in the text, attempts could be made to identify consensus sequences so the results can be put in better context with the existing literature. Uploading the consensus sequences as a Supplementary Table could also help with cross study comparisons.

Authors response –

As suggested by the author, we have now undertaken additional procedures in an attempt to improved identification of unassigned consensus regions. This was achieved using STIRRUPS, a vaginal 16S rDNA Reference Database that employs the USEARCH algorithm with a curated reference database for species-level classification of 16S rDNA partial sequence as described earlier in response to Reviewer 2. This enabled us to assign an additional 10 OTUs, 2 of which were present in the top 20 most abundant taxa observed in the sample cohort. We have now updated the results and text throughout the manuscript to include the new classifications. We have also made all raw sequencing data publicly available so that cross study comparisons can readily be made through reanalysis of our datasets.

g) Line 452: Vaginal microbiota composition was classified initially into two groups at genus level according relative abundance of *Lactobacillus*. Describe how *Lactobacillus* dominance was defined here.

Authors response – *Lactobacillus* dominance was based upon hierarchical clustering analysis using genera level taxonomy, with dominant samples having $\geq 81.6\%$ relative abundance. *Lactobacillus* deplete samples were defined as $< 54.2\%$ *Lactobacillus* abundance.

We have now updated the results section appropriately:

*‘Hierarchical clustering of genus level data for the whole cohort identified two major groups; those with $\geq 81.6\%$ *Lactobacillus* relative abundance, which were categorised as *Lactobacillus*-dominant and those with 54.2% *Lactobacillus* content, categorized as *Lactobacillus*-depleted communities. These were observed in 65.5% (57/87) and 34.5% (30/87) of samples at baseline respectively (Figure 2).’*

h) Line 454: Species-level data was used to classify samples into groups analogous to previously described vaginal community state types (CSTs) I-V. Describe how CSTs were assigned here.

Authors response – CSTs were defined on the basis of hierarchical clustering of relative abundance data from species level taxonomy data. This resulted in the identification of CSTs whereby CST I was characterised by *L. crispatus* dominance ($\geq 54.4\%$ relative abundance) and CST III by *Lactobacillus iners* dominance ($\geq 63.3\%$) content. Samples with less than 42.9% relative abundance of any *Lactobacillus* spp. was classified as CST IV.

The results section has been amended accordingly:
'Similar analyses were performed at species level with hierarchical clustering analysis identifying three of the previously described CST's; *CST I* classified as $\geq 54.4\%$ *Lactobacillus crispatus* content, *CST III* as $\geq 63.3\%$ *Lactobacillus iners* content. Anything with less than 42.9% *Lactobacillus* spp. content was classified as *CST IV*.'

3) Data accessibility

a) The SRA study has been registered, but not yet released. I assume the study will be released prior to publication.

Authors response –Yes, all raw sequence data has been registered to the SRA and will be made publicly available prior to publication.

b) Was consent given to permit any of the clinical fields to be openly released with the data, either in the supplement or in the metadata associated with the sequences?

Authors response –
All of the raw data (fastq files) associated to this project has been uploaded to the SRA along with metadata to facilitate reproducibility of the findings presented in our study.

c) Will other clinical phenotype data also be made available (controlled-access) in a database such as dbGAP?

Authors response –
As above, relevant metadata to facilitate reproducibility of the findings presented in our study have been made available in the SRA. Additional clinical phenotype data has not been made available but interested parties can contact the authors.

Reviewer 3

The manuscript is well written, and I really enjoyed reading the paper. Microbial communities may play an important role in the natural history of HPV infection and the cervical carcinogenesis. As the authors concluded, in the future, the use of vaginal microbiota analysis might be useful for the triage of patients in evaluating the CIN disease severity and determining the treatment.

The study population of the present study was quite small but because the cohort consisted of women with only conservatively treated CIN2, the size of the cohort and longitudinal study setting is respectable. The study population included also women from various ethnical groups which is an important issue regarding vaginal microbiome studies. The background characteristics were well described and surprisingly similar between the regression and non-regression groups. However, I think some of the background variables should have been considered as confounders and adjusted after all. Additionally, the composition of vaginal microbiome was not determined according to the background variables (at least the results were not presented).

Many thanks for your comment. We have addressed these issues as raised by previously reviewers and accordingly performed new analyses (see Table 2, Figure 3 and manuscript Results and Methods).

I have some minor comments to the manuscript:

Abstract:

- **The age of the participants (16-26 years) should be mentioned.**

Authors response –Thank you, this has now been added.

'We extracted and sequenced bacterial DNA from serially-collected vaginal samples from a cohort of 87 young women aged 16-26 years with histologically-confirmed, untreated CIN2 lesions to determine whether VMB composition affects rates of regression over 24 months.'

Methods:

- **The clinical definition of CIN2 regression should be described better.**

Authors response –Thank you, we have now amended the definition of CIN2 regression to read as follows:

We have amended the Methods sections accordingly:

'Regression of CIN2 was defined as negative cytology and/or negative biopsy in two consecutive visits and no further cytological or histological abnormality during follow-up. Histology was used in preference to cytology; if not available, cytology was used.'

- **At which time points were the biopsies (1-6?) taken?**

Authors response –A biopsy was taken at entry as a mandatory step for study enrolment to confirm CIN2 diagnosis. Biopsies during follow-up were performed on clinical grounds. Biopsy results, if available, were used in preference to cytology. If histology was not available, cytology was used.

We amended the text at 2 sections of the Methods to clarify this:

'Women between the ages of 16 to 26 years of age with histologically proven CIN2 at entry to the study were recruited'.

'A mandatory biopsy was performed at the first visit to confirm CIN2 for study entry. At subsequent visits, a cytology sample was collected and in addition, a colposcopy was performed. During follow-up, biopsies were taken on clinical grounds ie. suspicion of progression based on colposcopy. An exit biopsy was also performed in all patients who attended their final visit and gave consent regardless of clinical need. If the last visit was missed and no histology was available, cytology was used.'

- **The cohort has been collected between 2002 and 2007 – has the CIN diagnostics changed over time since then?**

Authors response – Despite advances and novel biomarkers, there has been no major change in the histological classification and diagnostics of CIN grades that would affect the results of this analysis.

- **Sequencing methods were appropriate and well described – however, the number of identified taxa (altogether 160) seems to be surprisingly high.**

Authors response –

Thank you for this comment. As pointed out by the reviewer, despite there being a high number of detected taxa, the majority of these were of low abundance and low frequency. This is consistent with ours and other studies of similar cohorts in the field (Mitra *et al.* Sci Rep 2015; Brotman *et al.* J Inf Dis 2014, Oh *et al.* Clin Microbiol Infect. 2015). As described in response to Reviewer 2, we have now limited our analyses to the top 20 most abundant species, which collectively account for >97% of total sequence reads. This has not altered the major findings of our study.

Results:

• **Figure 1: Those 10 patients who were not included in the sub-group analysis of non-regressors (one who progressed to CIN3 and nine who lost the follow-up), should be marked to the flowchart.**

Authors response – Thank you, this has been amended.

• **The word ‘deplete’ should be corrected as ‘depleted’ throughout the manuscript.**

Authors response – Thank you, this has been corrected.

• **Table 1:**

1. **P-values should be presented in a separate column.**
2. **It is not clear to me what ‘other ethnicity’ represents?**
3. **Because the number of sexual partners is not presented as nominal variable in the table, n/N (%) is not needed.**
4. **Why contraceptive injection (=progestin injection) is not included in the hormonal contraception group as well?**
5. **P-values should be presented separately by the presence of each genital infection.**

Authors response –

1. This has been amended.
2. Ethnicity was self-reported as either Black, Caucasian, Hispanic or Other in the collected demographics of this cohort. This includes mixed ethnicity groups etc.
3. This has been amended
4. We have clarified that by hormonal contraception we refer to oral combined (estrogen and progesterone) hormonal contraception.
5. This information has been added

• **Figure 2: This is very nice picture but showing bacterial species between genus and CST lines is not informative as the lines are barely visible. Again, I don’t understand why depot progestin injection has not been included to the group of hormonal contraceptives. There was a typo in the figure legend: hierarchial. I think also that the statement ‘there was no significant difference in variables that may impact on VMB composition etc.’ should be found also in the manuscript text, not only in the figure legend.**

Authors response – Thank you. We have now adjusted Figure 2 so that labelled bacterial genera are limited to the top 20 most abundant taxa. This has substantially improved the presentation and clarity of the data. We have amended the spelling error and added the information in at Line 175. Oral hormonal contraception and injectable hormonal contraception were collected separately in the database and presented as separate groups. There has been substantial literature on the impact of oral combined (with estrogen and progesterone) hormonal contraception on HPV persistence and CIN progression, which is why this was presented separately.

• **The sentence ‘CST IV was significantly associated non-regression compared with women with CST I etc...’ should be respelled. Especially numbers in parenthesis are not clearly presented in the current form.**

Authors response – Thank you, this has now been rectified and we have attempted to make the numbers in parentheses clearer.

‘CST IV was significantly associated with non-regression compared with women with CST I or CST III at baseline (non-regression rate - CST I: 8/21, 40.9% vs CST III: 16/36, 44.4% vs CST IV: 21/30, 69%; p=0.0420).’

- **Figure 3 is very nice and informative. There is a small typo in the figure legend: Outcomes at 12 months follow-up (or 12-month follow-up)**

Authors response – Thank you, this has been corrected

- **Figure 6 would be clearer, if pre- and post- prefixes were presented in the upper corner of figures A and B. Also, the prefix Lactobacillus should be added to -DEPLETED and -DOMINANT.**

Authors response – Thank you, this has been amended.

Discussion:

- **Discussion was very coherent, logical and easy to read. The references were topical, but I am wondering if all the sited references (altogether 53) are needed.**

Authors response – We agree that there are a lot of references, but we wanted to ensure that we gave adequate credit to the large number of authors who have contributed to the rapid acquisition of important data in this fast-advancing field. We are happy to delete some of the references if the editorial board feels that this is an excessive number of references.

REVIEWERS' COMMENTS:

Reviewer #1 (Remarks to the Author):

The authors have addressed the comments satisfactorily.

Reviewer #3 (Remarks to the Author):

The authors have improved the manuscript according to the comments and remarks. The paper has now enhanced remarkably, and I would pleasantly recommend it for the publication.

Reviewer #4 (Remarks to the Author):

Thanks for addressing the concerns/comments from Reviewer #2. I believe the authors' revisions are satisfactory regarding the reviewers' original comments. I have only one comment. The authors might have missed one important question raised by Reviewer #2 regarding Data Accessibility and Consent. The authors should address this question and this information should be included in the Methods Section.

b) Was consent given to permit any of the clinical fields to be openly released with the data, either in the supplement or in the metadata associated with the sequences?

REVIEWERS' COMMENTS:

Reviewer #1 (Remarks to the Author):

The authors have addressed the comments satisfactorily.

Author response: Thank you

Reviewer #3 (Remarks to the Author):

The authors have improved the manuscript according to the comments and remarks. The paper has now enhanced remarkably, and I would pleasantly recommend it for the publication.

Author response: Thank you

Reviewer #4 (Remarks to the Author):

Thanks for addressing the concerns/comments from Reviewer #2. I believe the authors' revisions are satisfactory regarding the reviewers' original comments. I have only one comment. The authors might have missed one important question raised by Reviewer #2 regarding Data Accessibility and Consent. The authors should address this question and this information should be included in the Methods Section.

b) Was consent given to permit any of the clinical fields to be openly released with the data, either in the supplement or in the metadata associated with the sequences?

Author response: Thank you. Regarding Data Accessibility and Consent, at the time of recruitment we did not seek explicit permission to openly release all clinical data in a data repository but we stated to the research group that we would maintain their confidentiality and anonymity. We have therefore uploaded only the basic clinical data (regression/persistence and HPV status (given that this was performed as a research test and not part of their routine clinical management)) alongside sequence data in order to ensure that patients remain unidentifiable. Further data is available upon request from interested parties.

We have included this information in the 'Data Availability' section. We have amended the Methods Section to contain the following sentence:

"All patients gave written informed consent to participate, and to maintain anonymity only basic clinical data has been released alongside sequence data."